# MMBench-Live: A Continuously Evolving Benchmark for Multimodal Models

Yuanzhi Liu [1]   Shousheng Zhao [1]   Bo Zhou [1]   Kongming Liang [1]   Zhanyu Ma [1]

## Abstract

Evaluation benchmarks are essential for assessing vision–language models (VLMs), but most multimodal benchmarks are static, making them vulnerable to temporal staleness, data contamination, and costly maintenance. We present MMBench-Live, a continuously evolving multimodal benchmark built by a multi-agent-driven automated pipeline. Our framework treats benchmark evolution as task-guided dataset construction, integrating structured benchmark specification, feedback-controlled real-time data acquisition, and verifiable QA generation with executable reasoning. To maintain cross-version comparability, we introduce a distribution-consistent update strategy that extracts task-related visual patterns from the original benchmark to guide data collection and filtering. Instantiated from MMBench, MMBench-Live contains 5.9K newly generated evaluation instances with a high answer correctness rate, while each update costs about $30 and takes 1–2 hours. Extensive evaluations show that MMBench-Live preserves stable model rankings, maintains semantic alignment with the original benchmark, and exhibits weaker contamination-related memorization signals, suggesting a practical and scalable paradigm for sustainable multimodal benchmark evolution. The project is available at https://github.com/PRIS-CV/MMBench-Live.

## 1. Introduction

Most existing vision-language evaluation benchmarks (Lu et al., 2022; Fu et al., 2023; Li et al., 2024; Liu et al., 2024b) adopt a static evaluation paradigm, implicitly assuming that a fixed test set can serve as a long-term proxy for model capability. This assumption becomes increasingly fragile as modern vision-language models (VLMs) scale rapidly and are trained on ever-growing web-scale corpora. Static benchmarks are vulnerable to temporal staleness, data contamination (Touvron et al., 2023a; Chen et al., 2024; Song et al., 2025b), and high maintenance costs, making frequent updates impractical at scale. However, dynamically updating a benchmark is not simply a matter of replacing old samples with new ones: a reliable live benchmark must introduce temporally fresh instances while preserving the original task semantics, capability coverage, data characteristics, and cross-version comparability.

Prior work has explored several strategies for updating multimodal evaluation datasets. Self-evolving methods based on visual perturbations or linguistic rewrites (Yang et al., 2025b; Zhang et al., 2026) may suffer from semantic drift and degraded comparability, while generation-based methods (Wang et al., 2026; Zhang et al., 2025a; Wen et al., 2025) can introduce distributional gaps under complex semantic constraints. Real-world data updating (Jiang et al., 2025; Shabtay et al., 2025; Jain et al., 2025; White et al., 2025) is a promising direction because it naturally improves temporal freshness and reduces direct reuse of static benchmark samples. Nevertheless, existing pipelines are often tied to fixed data sources, predefined task formats, or task-specific construction procedures. Consequently, continuously incorporating fresh real-world data across diverse multimodal tasks while preserving semantic coherence, distributional alignment, and evaluation stability remains challenging.

To address this challenge, we propose MMBench-Live, a continuously evolving multimodal benchmark built through a multi-agent-driven automated pipeline. Our framework models benchmark evolution as task-guided dataset construction. Starting from the original benchmark, it converts evaluation objectives, task hierarchy, and atomic tasks into structured benchmark descriptions, which serve as the semantic basis for data acquisition and QA generation. To preserve cross-version comparability, MMBench-Live identifies task-related visual patterns from the original benchmark and uses them to guide feedback-controlled real-time data acquisition, enabling the pipeline to refine poorly aligned retrieval queries and filter task-irrelevant candidates. For each collected image, the QA generation stage constructs a question, an answer, and an executable solution plan, which is verified through tool-supported reasoning to improve the reliability of automatically generated evaluation instances.

---

[1]Beijing University of Posts and Telecommunications, Beijing, China. Correspondence to: Kongming Liang <liangkongming@bupt.edu.cn>.

*Proceedings of the 43rd International Conference on Machine Learning*, Seoul, South Korea. PMLR 306, 2026. Copyright 2026 by the author(s).

We instantiate this framework by systematically updating MMBench (Liu et al., 2024b). The resulting MMBench-Live contains 5.9K newly generated evaluation instances, achieves a manual answer correctness rate of 96.06%, and completes each update within approximately 1–2 hours at a cost of about $30. Experiments on representative open-source VLMs show that MMBench-Live maintains stable cross-version model rankings, preserves the core semantic and distributional characteristics of the original benchmark, and exhibits weaker PaCoST-based (Zhang et al., 2024) memorization signals than MMBench. These results demonstrate that MMBench-Live provides a practical, scalable, and low-cost paradigm for sustainable multimodal benchmark evolution.

The main contributions can be summarized as follows:

- We introduce MMBench-Live, a continuously evolving multimodal benchmark constructed through a multi-agent-driven automated pipeline, enabling scalable and cost-efficient benchmark updates with temporally fresh real-world data.

- We propose a distribution-consistent benchmark updating strategy that combines task-related visual pattern identification with feedback-controlled data acquisition, preserving task semantics and visual characteristics across benchmark versions.

- We conduct systematic evaluations on MMBench-Live, showing high QA correctness, low construction cost, stable cross-version consistency, distributional alignment, and weaker contamination-related memorization signals.

## 2. Related Work

### 2.1. Data Contamination in Model Evaluation

Data contamination has become a critical issue in evaluating large language models (LLMs) and vision-language models (LVLMs), as overlap between benchmarks and pretraining corpora can substantially inflate reported performance. Prior studies show that such leakage is widespread even in commonly used benchmarks; for example, the LLaMA-2 report finds that over 16% of MMLU samples are contaminated, with a non-trivial fraction exhibiting severe leakage (Touvron et al., 2023b). Controlled experiments further demonstrate that even limited contamination can cause significant score inflation, disproportionately favoring larger models (Kocyigit et al., 2025). Position papers argue that these effects compromise the validity of model comparisons and lead to misleading conclusions about model capabilities (Sainz et al., 2023). The issue is exacerbated in multimodal evaluation, where leakage in either textual or visual modalities can artificially boost performance on

vision-language benchmarks (Song et al., 2025a). Together, these findings underscore the vulnerability of static benchmarks as pretraining corpora scale, motivating recent efforts toward automated and dynamic evaluation datasets designed to mitigate contamination (Wu et al., 2025).

### 2.2. Automatic Dataset Construction

Automatic dataset construction has gained increasing attention as manual curation and annotation limit the scalability of training and evaluation corpora. A major research direction frames dataset construction as automated instruction synthesis, where large language models generate and filter new instances from small seed sets, including reasoning-oriented and curriculum-based variants (Wang et al., 2023; Bu et al., 2025; Xu et al., 2024). These principles have been extended to multimodal settings, enabling large-scale vision-language instruction data synthesis with minimal human annotation, as exemplified by LLaVA (Liu et al., 2023b). Complementary approaches further incorporate programmatic structures and external artifacts—such as tool specifications, executable APIs, and web-scale data collection with lightweight filtering—to improve reliability and reduce manual effort (Guo et al., 2025; Xu et al., 2023; Liu et al., 2024a). However, most existing methods rely on post-hoc filtering rather than systematic validation, motivating closed-loop construction pipelines that explicitly enforce data correctness and evaluation relevance.

## 3. MMBench-Live

We construct MMBench-Live by converting the original MMBench into structured descriptions, acquiring new data in a task-aware manner, and generating executable and verified QA pairs for evaluation. Therefore, it can maintain (i) capability coverage consistency, (ii) distributional alignment with the original benchmark, and (iii) evaluation discriminativeness over time.

### 3.1. Benchmark Summary

To enable controlled and stable benchmark updates, we convert MMBench into a structured description, providing a unified representation for subsequent data acquisition and instance construction. Inspired by the prior works (Reuel et al., 2024; Sokol et al., 2025), we define a structured benchmark template for MMBench, including four components:

(1) **Evaluation purpose.** The evaluation purpose specifies the target capabilities that the benchmark is designed to assess. It provides a high-level semantic characterization of what competencies are being evaluated, such as visual perception, cross-modal reasoning, spatial understanding, or compositional inference.

(2) **Evaluation format.** The evaluation format specifies

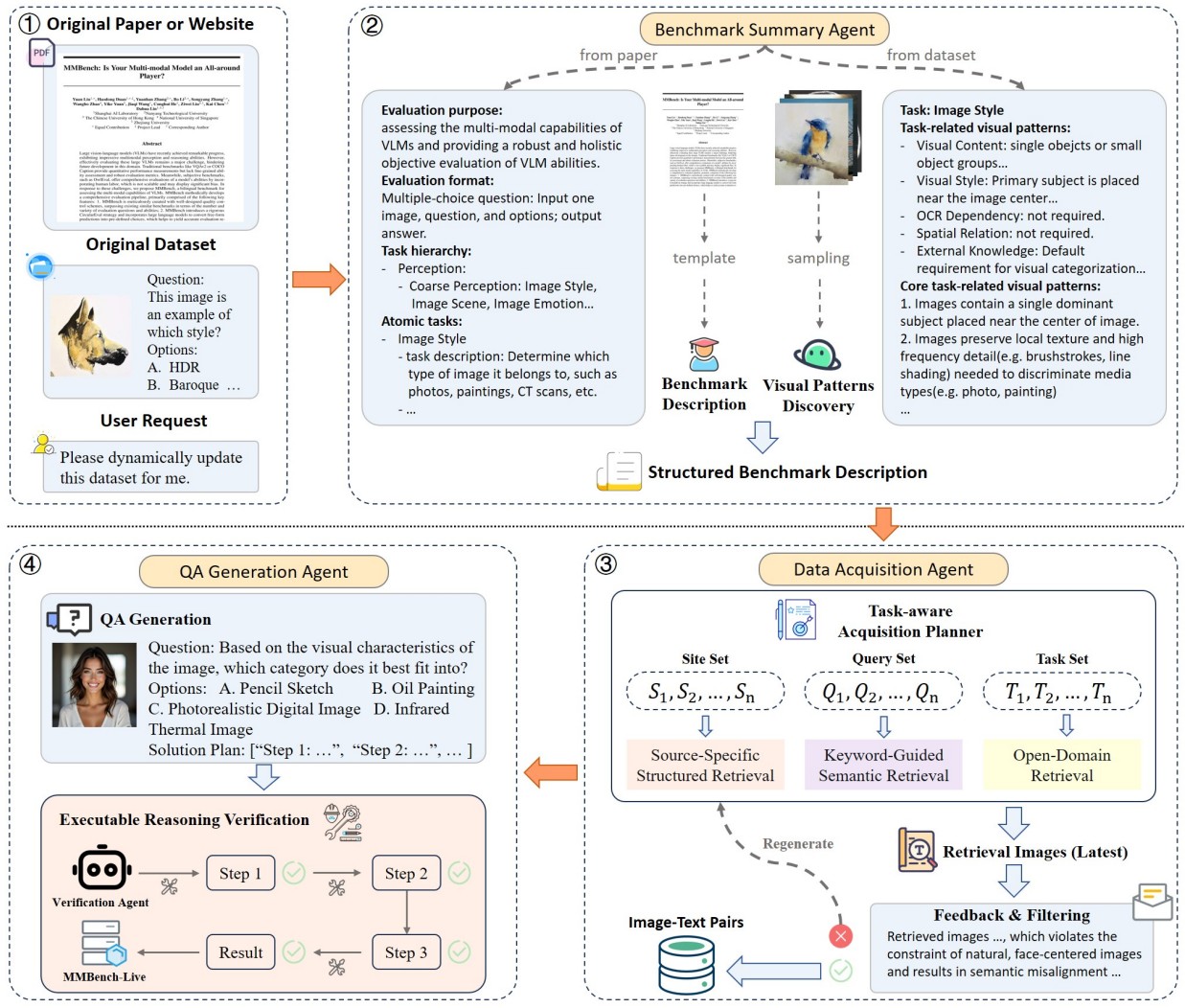

*Figure 1.* Overview of the MMBench-Live framework. MMBench-Live is constructed by converting the original MMBench into structured descriptions, acquiring new data in a task-aware manner, and generating executable and verified QA pairs for evaluation.

the structural form of an evaluation instance, including the modalities involved, the input–output interface, and the expected response type. An evaluation instance can be a single image paired with a multiple-choice question, an open-ended textual answer, or a structured output.

(3) **Task hierarchy.** The task hierarchy organizes the benchmark into a multi-level structure of tasks with various granularity. At the top level, the hierarchy captures coarse-grained evaluation tasks, which are progressively decomposed into finer-grained sub-tasks that correspond to specific and well-scoped capability requirements. For example, in MMBench, high-level *perception* tasks are further refined into fine-grained tasks such as *OCR* and *attribute recognition*.

(4) **Atomic tasks.** Atomic tasks correspond to the leaf-level units in the task hierarchy and define the smallest operational evaluation units. For each atomic task, we also define

a structured task description template consisting of three components: *Task purpose*, *Data sources and construction procedures* and *Task characterization*.

Task characterization is decomposed into textual and visual components. The textual aspect is represented by QA examples, which clarify the intended evaluation semantics. The visual aspect is captured by task-related visual patterns, defined as recurrent visual or semantic attributes that occur frequently within an atomic task but are not explicitly specified in the task definition, implicitly constraining valid evaluation instances.

To identify these patterns, we adopt a data-driven procedure summarized in Algorithm 1. For each atomic task $t$, we sample a subset of original instances and analyze them from five semantic perspectives inspired by MMBench: visual content, visual style, OCR dependency, spatial relations,

---

**Algorithm 1** Task-Related Visual Pattern Identification

---

1: **Input:** evaluation instances $E_t$ for atomic task $t$; sampling ratio $\rho$; analysis perspectives $P$
2: **Output:** task-related visual pattern set $B_t$
3: $E_t' \leftarrow \text{Sample}(E_t, \rho)$
4: $C_t \leftarrow \emptyset$
5: **for** $p \in P$ **do**
6:     $C_p \leftarrow \text{VLM\_Extract}(E_t', p)$
7:     $C_t \leftarrow C_t \cup C_p$
8: **end for**
9: $B_t \leftarrow \text{LLM\_Consolidate}(C_t)$
10: **return** $B_t$

---

and external knowledge dependency. A VLM extracts recurring patterns $C_t$ under each perspective, which are then consolidated by an LLM to merge similar patterns and remove redundancy, yielding a compact set of task-related visual patterns. These patterns guide subsequent data acquisition and filtering, ensuring consistency with the visual characteristics of the original evaluation data.

### 3.2. Task-aware Data Acquisition

This stage collects new evaluation data by converting each atomic task into an executable acquisition procedure. Rather than issuing generic search queries, the acquisition process is constrained by the task intent, required visual evidence, and available acquisition context. It consists of three components: an acquisition planner, an acquisition executor, and a feedback controller.

**Acquisition Planner.** For each atomic task $t$, the acquisition planner analyzes the structured task description $S_t$ and produces an acquisition plan:

$$P_t = \langle F_t, \ A_t, \ Q_t \rangle. \quad (1)$$

Here, $F_t$ denotes the observable fields that must be preserved with each collected image, such as titles, surrounding textual context, source metadata, or timestamps. $A_t$ denotes the acquisition strategy, and $Q_t$ denotes the generated query set when query-based retrieval is required.

**Acquisition Executor.** The Acquisition executor performs data collection according to the plan generated by acquisition planner. It supports three executable retrieval modes:

(1) Source-specific Structured Retrieval. This mode conducts structured crawling over designated websites specified in the acquisition plan. Relevant image links and associated textual fields are extracted from the crawler outputs and retained. This mode is primarily used when authoritative or task-specific data sources are explicitly defined.

(2) Keyword-Guided Semantic Retrieval. This mode submits a set of query keywords to the Google Image API to retrieve candidate images. To maintain temporal relevance and reduce potential contamination, retrieval is restricted to images uploaded within a one-year time window, and associated metadata are retained.

(3) Open-Domain Retrieval. This mode continuously samples recently uploaded public images from open-domain content streams via the Flickr API, together with their associated metadata. This mode does not rely on predefined sources or explicit semantic constraints, enabling broad coverage of diverse and naturally occurring visual content.

**Feedback Controller.** During data acquisition, retrieval queries may return visually plausible but task-irrelevant images, especially when the task involves fine-grained visual cues, implicit constraints, or subjective concepts. To maintain semantic alignment, we introduce a lightweight feedback controller that verifies retrieved candidates and refines poorly aligned queries during retrieval. The controller is guided by the task purpose $G_t$ and the task-related visual patterns $B_t$, which jointly specify the intended semantics and expected visual evidence of valid samples for task $t$.

For each retrieved candidate image $x_i$ with its associated textual context $c_i$, we instantiate a prompt-guided multimodal judge $J$ to evaluate whether the candidate is visually consistent with the task requirements:

$$(a_i, r_i) = J(x_i, c_i \mid G_t, B_t), \quad (2)$$

where $J$ denotes a multimodal judging model guided by task-specific prompts. Given the candidate image $x_i$, its textual context $c_i$, the task purpose $G_t$, and the task-related visual patterns $B_t$, the prompt instructs $J$ to check task consistency based on observable visual evidence and to output a binary acceptance decision $a_i \in \{0, 1\}$ together with one to three concise visual rationales $r_i$. These rationales are grounded in observable image evidence, such as object presence, scene type, visual attributes, action states, spatial relations, or other task-specific visual cues.

For each retrieval query, the controller examines the top-$K$ returned candidates. In our implementation, we set $K = 10$. The acceptance rate of the current query is computed as

$$\rho_q = \frac{1}{K} \sum_{i=1}^{K} a_i. \quad (3)$$

If $\rho_{q_m} \geq \tau$, the current query is considered sufficiently aligned and is retained for subsequent acquisition. Otherwise, the controller treats the rejected candidates as diagnostic examples. Their visual rationales are aggregated into a compact feedback signal $r$, which conditions the query update:

$$q_{m+1} = U(q_m \mid G_t, B_t, r), \quad \text{if } \rho_{q_m} < \tau. \quad (4)$$

**Algorithm 2** Feedback-controlled data acquisition

**Require:** $G_t$, $B_t$, $q_0$, $J$, $K = 10$, $\tau = 0.7$, $M = 3$
**Ensure:** Filtered candidate set $\mathcal{D}^+$
1: **for** $m = 0$ to $M - 1$ **do**
2:     $\mathcal{D}_m \leftarrow \text{Retrieve}(q_m, K)$
3:     $\{(a_i, r_i)\}_{i=1}^K \leftarrow \{J(x_i, c_i \mid G_t, B_t)\}_{(x_i,c_i)\in\mathcal{D}_m}$
4:     $\rho_{q_m} \leftarrow \frac{1}{K}\sum_{i=1}^K a_i$
5:     **if** $\rho_{q_m} \geq \tau$ **then**
6:         $q^\star \leftarrow q_m$
7:         **break**
8:     **end if**
9:     $r \leftarrow \text{Agg}(\{r_i \mid a_i = 0\})$
10:     $q_{m+1} \leftarrow U(q_m \mid G_t, B_t, r)$
11: **end for**
12: **if** $q^\star$ is undefined **then**
13:     $q^\star \leftarrow q_m$
14: **end if**
15: $\mathcal{D} \leftarrow \text{Retrieve}(q^\star)$
16: $\mathcal{D}^+ \leftarrow \{(x, c) \in \mathcal{D} \mid J(x, c \mid G_t, B_t) = 1\}$
17: **return** $\mathcal{D}^+$

---

The update operator $U$ is instantiated with a large language model. In this way, the updated query preserves the task intent while reducing the visual deviations observed in the previous retrieval round.

As summarized in Algorithm 2, this feedback process is repeated for at most three iterations. Once the query satisfies the acceptance criterion or reaches the maximum number of iterations, the controller stops updating the retrieval query. The final query is then used for subsequent candidate collection, while the judge is applied only as a candidate-level filter. This design allows the acquisition process to correct poorly aligned search queries early, while keeping the overall pipeline lightweight. It improves the semantic quality of collected samples without human annotation, model fine-tuning, or updates to the underlying judge model.

### 3.3. QA Generation and Verification

This stage constructs generate QA pairs from image data under a given task specification and verifies them through executable reasoning, ensuring that each evaluation instance is semantically aligned with the target task and admits a deterministic validation procedure.

**QA Generation.** Given an atomic task $t$ and an image $x$, the QA generator constructs a task-consistent and verifiable QA instance grounded in the image. To reduce systematic generation bias and improve generation quality, multiple multimodal large models are used to produce candidate QA pairs, and the best candidate is selected through model-based comparison and rule-based validity checks.

For each image, the generated instance is represented as a structured triplet:

$$z = (q, a, \pi), \tag{5}$$

where $q$ is the evaluation question, $a$ is the corresponding answer, and $\pi$ is an executable solution plan specifying how $a$ can be derived from the image. The solution plan provides an explicit reasoning procedure for subsequent verification through tool execution.

Each generated triplet is required to satisfy three criteria: consistency with the atomic task definition, explicit reliance on visual evidence from the image, and executability of the solution plan. This design turns QA generation from unconstrained question writing into structured instance construction, where the generated question, answer, and solution plan are jointly prepared for automatic validation.

**Executable Reasoning Verification** To address visual hallucinations and enhance the reliability of automatically generated QA pairs, we explicitly decouple correctness verification from direct vision-based model judgment and instead employ a text-driven executable reasoning process for validation.

Given a synthesized instance $(q, a, \pi)$, the solution plan $\pi$ is treated as a structured and interpretable sequence of executable reasoning operations. Specifically, we represent the plan as

$$\pi = (o_1, o_2, \ldots, o_K), \tag{6}$$

where $o_k$ denotes the executable operation at step $k$.

Importantly, the verification controller itself is vision-blind: it does not directly access the image but performs reasoning solely over the textual intermediate results generated by tool execution. Starting from the input image $x$, tools are executed sequentially according to the generated operation sequence to produce textual intermediate outputs:

$$z_k = T(o_k \mid x, z_{<k}), \quad k = 1, \ldots, K, \tag{7}$$

where $T$ denotes the tool execution process, and $z_{<k}$ represents previously obtained textual intermediate outputs.

The final verification outcome $\hat{a}$ is then predicted by a vision-blind language-model verifier conditioned only on the question and the tool-derived textual observations:

$$\hat{a} = \arg\max_y P(y \mid q, z_{1:K}). \tag{8}$$

The synthesized answer $a$ and the corresponding QA pair are accepted only if it matches the verified outcome $\hat{a}$,

$$\mathbb{I}(q, a, \pi) = \mathbb{I}[a = \hat{a}]. \tag{9}$$

Overall, this design constrains the verification process to explicit reasoning chains, thereby reducing reliance on implicit visual guessing and mitigating hallucinations.

*Table 1.* Results on MMBench-Live (L-2 abilities). Abbreviations adopted: LR for Logical Reasoning; AR for Attribute Reasoning; RR for Relation Reasoning; FP-C for Fine-grained Perception (Cross Instance); FP-S for Fine-grained Perception (Single Instance); CP for Coarse Perception.

| Model | Overall | CP | FP-S | FP-C | AR | LR | RR |
|---|---|---|---|---|---|---|---|
| DeepSeek-VL-7B-Chat (Lu et al., 2024) | 83.04% | **97.37%** | **75.37%** | 68.22% | **94.00%** | 67.17% | 85.60% |
| InstructBLIP-Vicuna-7B (Dai et al., 2023) | 73.85% | 92.05% | 60.14% | 55.66% | 91.73% | 56.95% | 73.44% |
| LLaVA-v1.5-7B (Liu et al., 2023a) | 74.83% | 91.56% | 66.14% | 56.80% | 86.98% | 59.13% | 76.98% |
| mPLUG-Owl2-7B (Ye et al., 2024) | 77.18% | 93.36% | 66.49% | 60.33% | 90.94% | 60.64% | 80.52% |
| Qwen3-VL-8B-Instruct (Team, 2025) | 80.41% | 80.36% | 72.76% | 70.72% | 88.11% | 72.36% | 84.42% |
| Qwen2.5-VL-7B-Instruct (Bai et al., 2025) | **83.10%** | 81.54% | **75.11%** | **77.47%** | 88.45% | **80.23%** | **85.71%** |

*Table 2.* Cross-version correlation of model performance between the original and updated benchmarks.

| Granularity | Pearson $r$ | Pearson $p$ | Spearman $\rho$ | Spearman $p$ |
|---|---|---|---|---|
| Task-Model Level | 0.5200 | $1.15 \times 10^{-9}$ | 0.4254 | $1.27 \times 10^{-6}$ |
| Model-Averaged Level | 0.8876 | 0.01823 | 0.8286 | 0.04156 |

## 4. Experiment

We evaluate the proposed framework through extensive experiments on MMBench-Live. Our analysis covers model performance, dataset quality, construction cost, distributional alignment, cross-version consistency, and data contamination, together with ablation studies examining the role of task constraints, semantic feedback, and underlying foundation models.

### 4.1. Evaluation Setup

**Size of Dataset.** Taking the MMBench (Liu et al., 2024b) dev set (4K) as the reference baseline, we apply our automated framework to construct MMBench-Live, which results in 5.9K newly generated evaluation QA pairs.

**Model Selected.** In the QA Generation stage, we employ a small pool of generation models, consisting of OpenAI GPT-5 Mini (OpenAI, 2025) and Gemini-3-Flash-Preview (Team et al., 2025), to construct QA pairs. All agent components are uniformly instantiated with OpenAI GPT-5 Mini as the underlying proxy model.

For executable reasoning verification, VisionReasoner (Liu et al., 2026) is used for object detection and segmentation, Depth Anything V2 (Yang et al., 2024) is used for monocular depth estimation, and a fine-tuned LLaVA (Liu et al., 2023a) model is used for visual attribute recognition. For text-related perception, Qwen2.5-VL (Bai et al., 2025) and GOT-OCR-2-HF (Wei et al., 2024) are used for OCR. Finally, Qwen3 (Yang et al., 2025a) is adopted as the text-only verifier, which aggregates the intermediate tool outputs and performs the final reasoning step.

For evaluation, we consider a diverse set of representative VLMs, including DeepSeek-VL-7B-Chat (Lu et al., 2024),

InstructBLIP-Vicuna-7B (Dai et al., 2023), mPLUG-Owl2-7B (Ye et al., 2024), LLaVA-v1.5-7B (Liu et al., 2023a), Qwen3-VL-8B-Instruct (Team, 2025), and Qwen2.5-VL-7B-Instruct (Bai et al., 2025). To guarantee fair and deterministic comparisons, the generation temperature of all evaluated models is fixed to zero during inference.

### 4.2. Evaluation Results on MMBench-Live

Table 1 summarizes the performance of different models in MMBench-Live. Overall, Qwen2.5-VL achieves the best aggregate performance, slightly outperforming DeepSeek-VL. In contrast, earlier models obtain substantially lower scores, indicating limited its generalization capability to continuously evolving benchmarks. A clear performance divergence is observed across evaluation dimensions. DeepSeek-VL excels in perception-oriented tasks, achieving the highest scores on coarse perception and attribute recognition, reflecting strong coarse-level visual understanding and attribute recognition ability. In comparison, Qwen2.5-VL consistently outperforms other models in fine-grained perception and reasoning-intensive dimensions, highlighting its advantage in detailed visual understanding and logical reasoning.

We also observe that Qwen2.5-VL outperforms the more recent Qwen3-VL across multiple evaluation dimensions. A similar performance reversal is also observed on MMBench. The performance degradation of Qwen3-VL is primarily concentrated in cross-instance fine-grained comparison and spatial reasoning tasks, where the model more frequently exhibits misaligned comparison targets and incorrect reference-frame binding.

As shown in Figure 2, the transition from MMBench to MMBench-Live does not induce global ranking instability.

Instead, ranking variations are largely driven by a small subset of highly discriminative sub-tasks, indicating that dynamic benchmark updates can surface fine-grained capability differences while preserving overall evaluation stability.

### 4.3. Dataset Quality and Construction Cost

**QA Correctness.** To evaluate the correctness of automatically constructed QA pairs, we conduct a manual verification study. For each task, 10% of the images are randomly sampled, and their corresponding QA pairs are manually verified. This procedure is repeated three times with independent samples to reduce variance. Across all tasks and trials, the QA pairs achieve an average answer correctness rate of 96.06%, demonstrating high annotation reliability. This statistic measures the correctness of the final answers only, rather than the validity of the executable solution plans. For failure cases, we provide a representative case analysis in Appendix A.

**Construction Cost.** We evaluate the construction cost from both monetary and time perspectives. Prior studies (Tang et al., 2024; Zhang et al., 2025b; Chen et al., 2025) on manual dataset construction report that, for a dataset at the scale of the 6K-sample MMBench-Live, manual pipelines typically incur costs on the order of thousands to tens of thousands of dollars, with turnaround times of approximately one to two months. In contrast, our automated pipeline completes a benchmark update within approximately 1–2 hours, with a total monetary cost of about $30, where QA generation accounts for the majority of the expense (around 60%). Overall, the proposed approach reduces both construction cost and turnaround time by multiple orders of magnitude, enabling frequent and sustainable benchmark updates.

### 4.4. Cross-Version Consistency of Model Performance

To examine whether model capabilities remain consistent across benchmark versions, we conduct a correlation-based analysis comparing model performance on the original benchmark and the dynamically updated benchmark, using Pearson correlation for linear agreement (Pearson, 1895) and Spearman rank correlation for relative ordering consistency (Spearman, 1904).

At the Task-Model Level, each model–capability pair is treated as an independent data point, enabling a fine-grained comparison of capability-level performance across benchmark versions. As shown in Table 2, we observe a stable positive correlation, indicating that models performing well on the original benchmark tend to maintain higher performance on the updated benchmark despite variability across individual capabilities.

At the Model-Averaged Level, performance is aggregated across all capability dimensions for each model before computing correlations. The resulting agreement is notably stronger, suggesting that the relative ordering of models at the overall capability level is largely preserved across benchmark versions. Together, these results indicate that the dynamic update maintains cross-version consistency.

### 4.5. Data Contamination Analysis

To assess potential contamination-related memorization effects in multimodal evaluation benchmarks, we adopt the PaCoST framework (Zhang et al., 2024). PaCoST compares model confidence between original questions and their meaning-preserving paraphrases, based on the assumption that memorized or leaked evaluation samples may induce higher confidence on the original formulation than on paraphrased variants.

It is important to note that our analysis should be interpreted as a proxy-based contamination-related signal rather than a direct diagnosis of multimodal data leakage. Since PaCoST measures confidence shifts between original and paraphrased questions, it is primarily sensitive to text-side memorization effects and cannot explicitly distinguish visual leakage, metadata overlap, or exposure through web content. Therefore, our goal is not to determine whether a benchmark is absolutely contaminated, but to compare the relative strength of memorization-related confidence shifts under the same evaluation protocol.

In our experiments, we use Qwen2.5-VL-7B-Instruct as the evaluated model and measure its confidence on both original and paraphrased questions. After producing a multiple-choice answer, the model provides a binary self-assessment of correctness. Based on the decoding probabilities during this decision, we define a confidence score $c \in [0, 1]$ that reflects the model's confidence in its predicted answer. For each sample $i$, we compute the confidence difference

$$\Delta_i = c_i^{\text{orig}} - c_i^{\text{para}}, \tag{10}$$

and conduct a one-sided paired $t$-test to evaluate whether $\mathbb{E}[\Delta] > 0$. We further report Cohen's $d$ as a standardized effect size to quantify the magnitude of confidence shift.

As shown in Table 3, both benchmarks exhibit statistically significant confidence shifts under the PaCoST criterion. However, MMBench-Live shows a smaller mean confidence difference and a lower effect size than MMBench. This indicates that, under the same PaCoST-based protocol, MMBench-Live exhibits a weaker memorization-related confidence bias than the original MMBench. We provide additional contamination-related analysis and comparisons with edited benchmark variants in Appendix B.

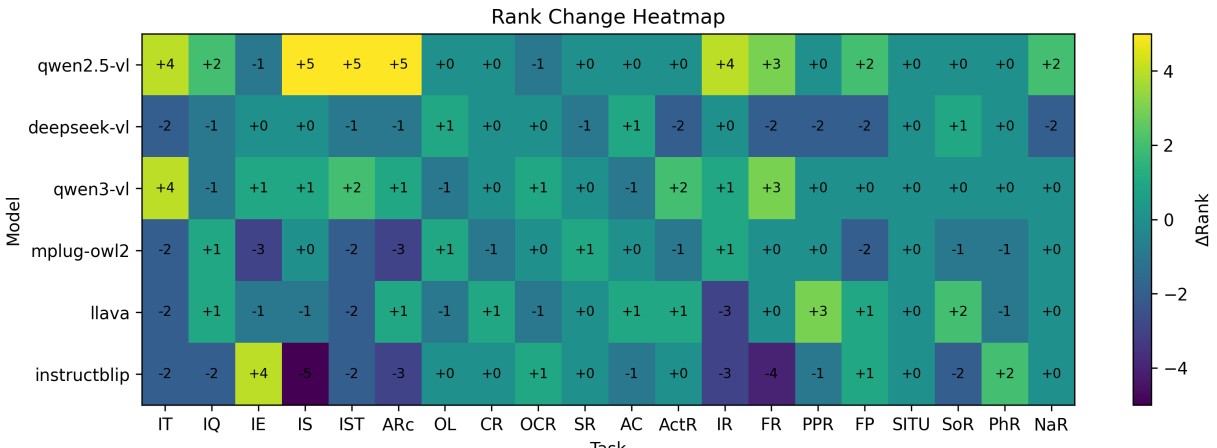

*Figure 2.* Rank change heatmap between MMBench and MMBench-Live. Each cell shows the rank difference $\Delta\text{Rank} = \text{RankLive} - \text{RankMMBench}$ for a given model–task pair. All tasks are denoted using abbreviated names and full task names are provided in the appendix C.

*Table 3.* Overall data contamination comparison between MMBench and MMBench-Live.

| Benchmark | Mean $\Delta$ | Std $\Delta$ | $t$-stat | $p$ (one-sided) | Cohen's $d$ |
|---|---|---|---|---|---|
| MMBench (Liu et al., 2024b) | 0.0118 | 0.1220 | 6.38 | $9.5 \times 10^{-11}$ | 0.097 |
| MMBench-Live | **0.0060** | **0.1016** | 4.54 | $2.9 \times 10^{-6}$ | **0.059** |

### 4.6. Task-Level Distribution Consistency

To further examine whether MMBench-Live preserves the task semantics of the original benchmark, we conduct a complementary task-level distribution analysis. For fairness and comparability, the sample ratio across tasks is explicitly balanced during construction. In addition, since MMBench is entirely composed of multiple-choice questions, MMBench-Live also uses multiple-choice QA pairs with the same number of answer options, ensuring that the answer format is controlled by design.

We analyze the spatial-relation subtask as a representative case, where task-related relation labels can be reliably recovered from QA pairs using an LLM. We compare both the label vocabulary and the frequency distribution between MMBench and MMBench-Live. Based on the recovered labels, MMBench contains 18 label types, with the top-5 labels being north, east, west, northwest, and south. In comparison, MMBench-Live contains 36 label types, with the top-5 labels being north, northeast, east, south, and west. The strong overlap among high-frequency labels indicates that the updated benchmark preserves the core semantic structure of this subtask while expanding label coverage.

We attribute this consistency to the task-related visual patterns and our distribution-consistent update strategy, which help the acquisition agent capture recurring structures in the original benchmark, such as the prevalence of map-based instances in spatial-relation questions.

### 4.7. Ablation Study and Analysis.

**Effect of Task-Related Visual Patterns and Semantic Feedback.** We analyze the role of task-related visual patterns and semantic feedback in preserving distributional alignment during dynamic dataset updates. We compare a pattern-aware update pipeline with an unconstrained variant under a matched-sample protocol, evaluating both against the original dataset. Distributional alignment is measured using FID computed on CLIP-ViT-L/14-336 image embeddings and KL divergence computed in a 256-dimensional PCA subspace, with FID averaged over 10 random trials.

As shown in Table 4, the pattern-aware update achieves lower FID and reduced variance compared to the unconstrained variant, indicating improved global semantic alignment. Beyond global alignment, PCA-based KL divergence reveals structural differences: with task-related visual patterns and semantic feedback enabled, $\text{KL}(A \parallel B)$ slightly exceeds $\text{KL}(B \parallel A)$, yielding a small positive $\Delta\text{KL}$ and suggesting that the updated data form a compact sub-distribution supported by the original dataset. In contrast, removing these mechanisms reverses this relation, resulting in a negative $\Delta\text{KL}$ that reflects reduced coverage and increased structural drift. Overall, these results highlight the importance of task-related visual patterns and semantic feedback in preserving distributional alignment during

*Table 4.* Distribution alignment between the dynamically updated dataset and the original dataset, where A denotes data collected by the dynamic update pipeline (with or without task-related visual patterns and semantic feedback), and B denotes the MMBench dataset.

| Updated Dataset | FID ↓ (mean±std) | PCA-KL(A‖B) ↓ | PCA-KL(B‖A) ↓ | ΔKL(A‖B − B‖A) |
|---|---|---|---|---|
| dataset (w/ visual patterns + feedback) | **0.1831 ± 0.0007** | 109.321 | **95.514** | **+13.807** |
| dataset (w/o visual patterns + feedback) | 0.2035 ± 0.0012 | **108.261** | 142.969 | **-34.708** |

*Table 5.* Results on MMBench-Live (L-2 abilities) on the Qwen-based version. Abbreviations adopted: LR for Logical Reasoning; AR for Attribute Reasoning; RR for Relation Reasoning; FP-C for Fine-grained Perception (Cross Instance); FP-S for Fine-grained Perception (Single Instance); CP for Coarse Perception.

| Model | Overall | CP | FP-S | FP-C | AR | LR | RR |
|---|---|---|---|---|---|---|---|
| DeepSeek-VL-7B-Chat (Lu et al., 2024) | 80.81% | **92.77%** | 68.50% | 69.60% | **88.47%** | 72.21% | 84.01% |
| InstructBLIP-Vicuna-7B (Dai et al., 2023) | 70.73% | 88.01% | 48.88% | 57.38% | 85.14% | 58.15% | 72.07% |
| LLaVA-v1.5-7B (Liu et al., 2023a) | 72.05% | 85.91% | 55.59% | 61.12% | 82.31% | 62.26% | 74.77% |
| mPLUG-Owl2-7B (Ye et al., 2024) | 74.30% | 88.01% | 58.35% | 60.90% | 86.31% | 65.69% | 76.35% |
| Qwen3-VL-8B-Instruct (Team, 2025) | **82.45%** | 78.62% | **72.72%** | **77.86%** | 84.75% | **80.45%** | **85.02%** |
| Qwen2.5-VL-7B-Instruct (Bai et al., 2025) | 79.97% | 78.55% | 69.79% | 72.91% | 82.80% | 76.67% | 84.91% |

dynamic benchmark updates.

**Does the Effectiveness of the Pipeline Depend on GPT?**
To assess whether the proposed pipeline depends on GPT-specific capabilities, we replace the large-model components in task specification and data acquisition with Qwen3 and Qwen3-VL. As shown in Table 5, the resulting MMBench-Live still exhibits clear performance separation across model families, indicating that the pipeline remains effective without relying on GPT models.

We further examine cross-version consistency between the original benchmark and the Qwen-based dynamic benchmark. Stable positive correlations are observed at both the Task-Model level (Pearson $r = 0.4574$, Spearman $\rho = 0.4408$) and the Model-Averaged level (Pearson $r = 0.9084$, Spearman $\rho = 0.8286$), suggesting that overall model ordering remains largely preserved. These results demonstrate that the proposed method can stably construct discriminative and cross-version-consistent benchmarks even when GPT models are fully replaced.

## 5. Limitations and Discussion

Despite its effectiveness in mitigating data contamination and improving evaluation timeliness, the proposed dynamic benchmark construction framework has limitations. Dynamic updates cannot fully eliminate implicit memorization of high-frequency visual concepts, particularly for recurring objects, scenes, or styles. Moreover, the quality of automatically constructed evaluation instances is bounded by the capabilities of the underlying agents and foundation models, and subtle noise may persist under visual ambiguity or underspecified constraints. From an extensibility perspective, the current task-grounded paradigm prioritizes cross-version

comparability and stable evaluation objectives, which facilitates tracking progress along predefined capabilities but limits task-space expansion. Future work will explore complementary task-expansive update strategies that introduce new task formulations alongside task-grounded updates to enable more adaptive evaluation of evolving VLMs.

## 6. Conclusion

This work revisits multimodal benchmark construction under the challenges of temporal obsolescence, data contamination, and high maintenance cost in static evaluation paradigms. We propose a task-centered, multi-agent framework that treats benchmark evolution as a task-guided dataset construction process, integrating structured benchmark descriptions, task-grounded data acquisition with feedback control, and automated instance construction with executable verification. Instantiated on MMBench, the resulting MMBench-Live enables fully automated updates with temporally recent and semantically aligned evaluation instances while preserving cross-version comparability. Experiments demonstrate high QA correctness, improved robustness to contamination, and stable performance correlations across versions. The proposed framework is general and extensible to other multimodal benchmarks, providing a practical paradigm for sustainable evaluation of evolving vision–language models.

## Impact Statement

This paper presents work whose goal is to advance the field of Machine Learning. There are many potential societal consequences of our work, none which we feel must be specifically highlighted here.

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

## A. Failure Analysis

Although executable reasoning verification substantially improves the reliability of automatically generated QA pairs, it does not eliminate all errors. We observe that the remaining failures are often not caused by low-level perception, but by the final reasoning stage that aggregates tool-derived evidence.

Figure 3 shows a representative failure case from a spatial-relation question about toy cars. The question asks: "Comparing the toy cars in the image, which statement regarding their colors and positions is correct?" One candidate option states that "There is a yellow car and an orange car in the same diagonal row starting from the left." In this case, the perception modules correctly identify the toy cars and the rotated grid layout. The upstream tools also return the relevant color attributes and bounding boxes. However, the final reasoning stage incorrectly concludes that the yellow and orange cars lie on the same diagonal.

A closer inspection shows that the error comes from interpreting the rotated grid structure. Under the rotated reference frame, the upper-right direction should be treated as a row direction rather than a diagonal direction. Therefore, the two cars are not on the same diagonal, even though their relative positions can appear diagonal under the original image coordinate system. This indicates that the main error source lies in relation reasoning over structured tool outputs, rather than in object detection, attribute recognition, or localization.

This case highlights both the benefit and limitation of executable verification. On the one hand, the tool chain makes the failure traceable by separating perception results from final reasoning. On the other hand, the verifier may still make incorrect relational inferences when the scene involves ambiguous reference frames, rotated layouts, or non-standard spatial structures. Future work may incorporate explicit geometric normalization or rule-based spatial consistency checks to further reduce such errors.

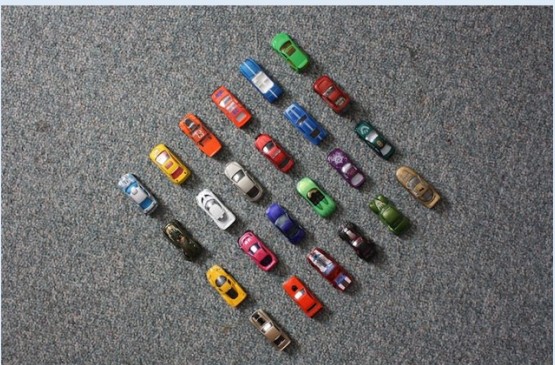

**Question:** Comparing the toy cars in the image, which statement regarding their colors and positions is correct?

**Options:**
A. There are two identical bright green cars placed next to each other.
B. The purple car is located directly above a light green car with black racing stripes.
C. There is a yellow car and an orange car in the same diagonal row starting from the left.
D. All the blue cars are clustered together in one corner of the arrangement.

**Verification:** image contains ~20 toy cars arranged in a rotated grid (diamond). Detect_and_seg provided bounding boxes for each car. see_attribute on two adjacent boxes returned: bbox [237,143,296,204] -> color: [\"orange\",...]; bbox [198,184,254,241] -> color: [\"yellow\",...]. The centers of these boxes (orange ≈ (266.5,173.5), yellow ≈ (226,212.5)) place the orange car up-right of the yellow car along the same diagonal, supporting Option C. Other options are not supported by the tool outputs: see_attribute found green cars at widely separated boxes (e.g. [360,36,414,91] and [394,229,449,284]) so Option A is false; there is no see_attribute evidence of a purple car directly above a light-green car with black racing stripes as described in Option B; blue cars detected by recognize/detect_and_seg are at multiple scattered boxes, not clustered in one corner (Option D false). Therefore Option C is correct based on the tool outputs.

*Figure 3.* A representative failure case of executable reasoning verification.

## B. Additional Contamination Analysis

We provide additional PaCoST-based contamination-related analysis to further compare MMBench-Live with VLB-style (Yang et al., 2025b) image editing variants. Here, Adding, Elimination, and Outpainting denote edited variants constructed by adding visual content, removing visual content, and extending image regions, respectively.

As shown in Table 6, the edited variants, including adding, elimination, and outpainting, exhibit substantially larger confidence shifts and effect sizes than MMBench-Live. In contrast, MMBench-Live shows a much smaller mean confidence shift and Cohen's $d$. Combined with temporal isolation in data acquisition, these results suggest that our live updating paradigm is more effective than image-editing-based variants in reducing contamination-related memorization signals.

We emphasize that PaCoST should be interpreted as a proxy-based analysis rather than a definitive contamination detector. Although the results provide suggestive evidence, they are insufficient on their own to draw strong conclusions about multimodal contamination, especially given the limited transparency of large-scale multimodal pretraining corpora.

*Table 6.* Additional PaCoST-based confidence-shift comparison between edited variants and MMBench-Live.

| Benchmark Variant | Mean $\Delta$ | Std $\Delta$ | $p$ (one-sided) | Cohen's $d$ |
|---|---|---|---|---|
| Adding (Yang et al., 2025b) | 0.042 | 0.187 | $6.1 \times 10^{-14}$ | 0.224 |
| Elimination (Yang et al., 2025b) | 0.038 | 0.200 | $3.7 \times 10^{-12}$ | 0.191 |
| Outpainting (Yang et al., 2025b) | 0.042 | 0.205 | $1.1 \times 10^{-13}$ | 0.205 |
| MMBench-Live | **0.006** | **0.102** | $2.9 \times 10^{-6}$ | **0.059** |

# C. Task Abbreviations

| Abbrev. | Task Name |
|---|---|
| IT | Image Topic |
| IQ | Image Quality |
| IE | Image Emotion |
| IS | Image Scene |
| IST | Image Style |
| ARc | Attribute Recognition |
| OL | Object Localization |
| CR | Celebrity Recognition |
| OCR | Optical Character Recognition |
| SR | Spatial Relationship |
| AC | Attribute Comparison |
| ActR | Action Recognition |
| IR | Identity Reasoning |
| FR | Function Reasoning |
| PPR | Physical Property Reasoning |
| FP | Future Prediction |
| SITU | Structuralized Image-Text Understanding |
| SoR | Social Relation |
| PhR | Physical Relation |
| NaR | Nature Relation |

*Table 7.* Task abbreviations and corresponding full task names.

## D. MMBench-Live Representative QA Examples

### D.1. QA Examples on the GPT-based Version

## Image Style

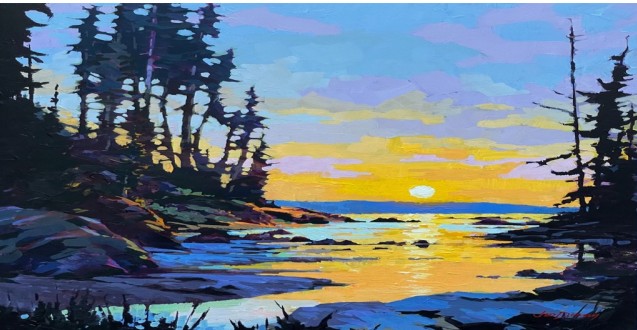

**Question:**
Based on the visual characteristics such as visible brushstrokes, texture, and color blending, what is the primary medium of this artwork?
A. Photography
B. Oil Painting
C. Pencil Sketch
D. 3D Render
**Answer:B**

**Solution Plan**

**Step 1:** Observe the overall texture and style of the image. The image displays prominent, thick applications of color (impasto technique) and visible brushstrokes.
**Step 2:** Evaluate the color transitions and blending. The colors are layered and blended in a way characteristic of physical media like oil or acrylic paint rather than digital gradients or photographic captures.
**Step 3:** Check for specific details such as the artist's signature in the bottom right corner ('Jed Dorsey'), which is typical for traditional paintings.
**Step 4:** Verify against metadata. The title 'How to Improve Your Paintings With Better Design!' and search query 'painting' support the conclusion that this is a painting, specifically an oil or acrylic painting based on the texture.

## Image Topic

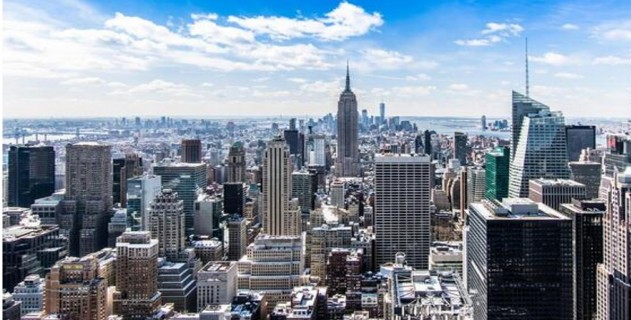

**Question:**
Which of the following categories best describes the subject and composition of this image?
A. A portrait of a person in an urban environment
B. A close-up macro shot of architectural details
C. A wide-angle cityscape showing a dense skyline
D. A text-heavy advertisement or infographic
**Answer:C**

**Solution Plan**

**Step 1:** Observe the overall composition of the image, noting the high vantage point and the vast number of buildings extending to the horizon.
**Step 2:** Identify key landmarks like the Empire State Building to confirm the setting is a major city (New York City).
**Step 3:** Evaluate the subject matter against the options: there are no prominent people (ruling out A), the view is expansive rather than a close-up (ruling out B), and there is no significant text (ruling out D).
**Step 4:** Verify that the visual evidence of a dense, wide-angle urban landscape matches the category 'cityscape'.

## Image Scene

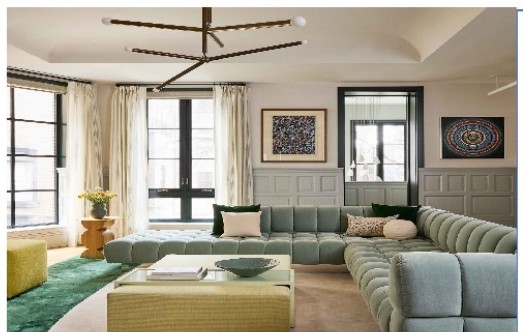

**Question:**
Based on the visual elements present in the image, such as the furniture arrangement and wall decor, what type of environment is depicted?
A. A public art gallery
B. A luxury hotel lobby
C. A furniture showroom
D. A residential living room
**Answer:D**

**Solution Plan**

**Step 1:** Identify the main furniture pieces in the image, which include a large modular sectional sofa, a coffee table with books and a bowl, an ottoman, and a side table with flowers.
**Step 2:** Observe the background and architectural features, such as the windows with curtains, wainscoting on the walls, and personalized wall art (framed abstract pieces).
**Step 3:** Analyze the overall atmosphere and layout. The intimate arrangement of cushions, a rug, and the presence of personal items like books suggest a private living space rather than a public or commercial setting.
**Step 4:** Verify the environment type against the options. While it has high-end decor, the specific combination of residential-style windows, curtains, and a domestic seating arrangement strongly indicates a residential living room.
**Step 5:** Reference the auxiliary metadata (title) which mentions 'Living Room Ideas' to support the visual conclusion.

## Image Emotion

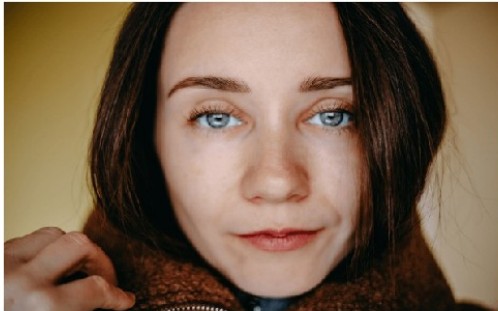

**Question:**
Based on the person's expression and the overall visual tone of the image, which emotion is most accurately conveyed?
A. Melancholy
B. Cheerful
C. Terrified
D. Enthusiastic
**Answer:A**

**Solution Plan**

**Step 1:** Analyze the facial expression of the person in the close-up portrait, focusing on the eyes and mouth.
**Step 2:** Observe the eyes, which appear glassy and slightly downcast, often associated with sadness or a somber mood.
**Step 3:** Evaluate the mouth, which is set in a neutral to slightly downturned position, lacking any smile or upward curve.
**Step 4:** Consider the overall lighting and color palette, which features muted, warm tones that contribute to a reflective or quiet atmosphere rather than an energetic one.
**Step 5:** Compare these observations with the provided options. 'Cheerful' and 'Enthusiastic' are contradicted by the lack of a smile. 'Terrified' is contradicted by the lack of wide eyes or a tense facial structure. 'Melancholy' aligns with the somber, quiet, and slightly sad expression observed.

# Image Quality

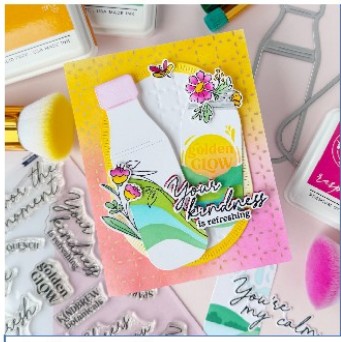

**Question:**
Which of the following best describes the lighting and exposure quality of this product photograph?
A. Well-lit and bright with high clarity and balanced exposure.
B. Underexposed and dark, making the colors look muddy.
C. Extremely high contrast with harsh shadows obscuring details.
D. Blurry and out of focus across the entire frame.
**Answer:A**

## Solution Plan

**Step 1**: Observe the overall brightness of the image. The white areas (like the ink pads and the white parts of the card) appear clean and bright without being 'blown out' (losing detail).
**Step 2**: Check for clarity and focus. The central greeting card, the text on the ink pads, and the stamps are sharp and easy to read, indicating good focus.
**Step 3**: Analyze the shadows. The shadows cast by the brushes and ink pads are soft and subtle, suggesting a diffused, high-quality light source typical of professional product photography.,
**Step 4**: Evaluate the color representation. The pinks, yellows, and greens are vibrant and well-saturated, which is a hallmark of balanced exposure.
**Step 5**: Compare these observations with the options. Option B accurately reflects these characteristics.

# Attribute Recognition

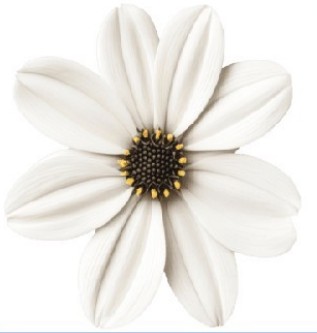

**Question:**
Based on the visual characteristics of the flower, which of the following best describes the shape and color of its petals?
A. Sharp, pointed petals that are bright yellow
B. Rounded, oblong petals that are primarily white
C. Small, circular petals that are dark green
D. Heart-shaped petals that are deep red
**Answer:B**

## Solution Plan

**Step 1:** Identify the main object in the image, which is a single flower.
**Step 2:** Observe the appearance characteristics of the petals, noting their shape and color.
**Step 3:** The petals are long and rounded at the ends (oblong) and their primary color is white with subtle textures.
**Step 4:** Compare these visual findings with the provided options to select the most accurate description.

# Celebrity Recognition

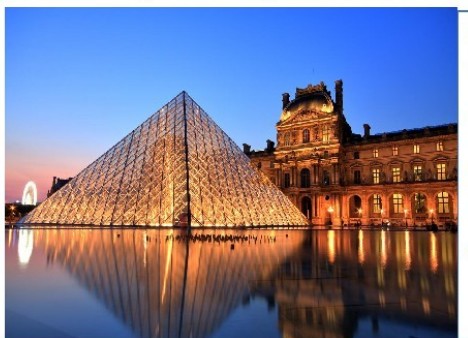

**Question:**
Which world-famous landmark is primarily featured in this image?
A. The British Museum
B. The Met Museum
C. The Guggenheim Museum
D. The Louvre Museum
**Answer:D**

## Solution Plan

**Step 1:** Identify the central, distinct architectural structure in the image, which is a large glass and metal pyramid.
**Step 2:** Observe the surrounding classical architecture and the reflecting pool, which are characteristic of the Cour Napoléon.
**Step 3:** Recognize the distant Ferris wheel (Roue de Paris) on the left, further contextualizing the location in Paris.
**Step 4:** Match these visual features to the well-known landmark, the Louvre Museum (Musée du Louvre) in Paris, France.
**Step 5:** Select the corresponding option (B).

# Object Localization

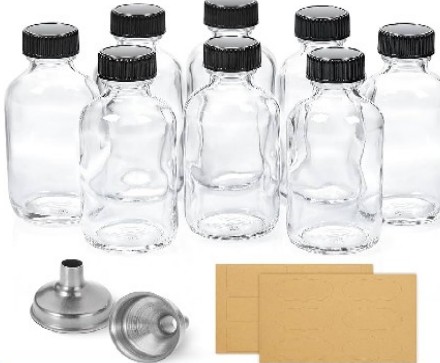

**Question:**
How many clear glass bottles with black lids are shown in the image?
A. 6
B. 8
C. 10
D. 12
**Answer:B**

## Solution Plan

**Step 1:** Identify the main group of objects which are clear glass bottles with black lids.
**Step 2:** Systematically count the bottles from left to right, front row to back row.
**Step 3:** In the front row, there are 3 bottles clearly visible.
**Step 4:** In the middle row, there are 3 bottles visible behind the front row.
**Step 5:** In the back row, there are 2 bottles visible towards the sides.
**Step 6:** Summing them up: 3 (front) + 3 (middle) + 2 (back) = 8 bottles."
**Step 7:** Verify this against the auxiliary title information which mentions an '8 Pack'.

## OCR

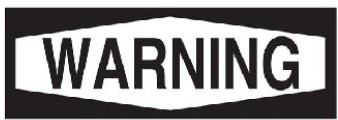

**Question:**
According to the text on this warning sign, what specific health risks are associated with the chemicals present in the area?
A. Respiratory illness and skin irritation
B. Vision loss and hearing impairment
C. Cancer, birth defects, or other reproductive harm
D. Chronic fatigue and digestive issues
**Answer:C**

**Solution Plan**

**Step 1:** Identify the main text block below the word 'WARNING' on the sign.
**Step 2:** Use optical character recognition (OCR) to read the specific message provided on the sign.
**Step 3:** Locate the phrase describing the effects of the chemicals: 'TO CAUSE CANCER, BIRTH DEFECTS, OR OTHER REPRODUCTIVE HARM'.
**Step 4:** Match this text to option C.

## Spatial Relationship

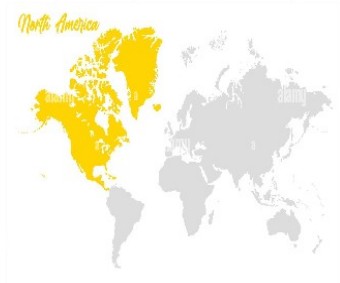

**Question:**
Based on the provided world map, what is the relative position of the continent highlighted in yellow compared to the rest of the world's landmasses shown in gray?
A. Northwest
B. Southeast
C. Southwest
D. Northeast
**Answer:A**

**Solution Plan**

**Step 1:** Identify the two main visual groups in the image based on color: the continent highlighted in yellow and the other landmasses shown in light gray.
**Step 2:** Read the text label in the top-left corner, which identifies the yellow region as 'North America'.
**Step 3:** Analyze the spatial arrangement of the landmasses on the map, assuming a standard north-up orientation.
**Step 4:** Observe that the yellow highlighted region (North America) is located in the upper (North) and leftmost (West) section of the map relative to the gray-colored landmasses of South America, Europe, Africa, Asia, and Australia.
**Step 5:** Conclude that the highlighted continent is in the Northwest position relative to the majority of the other landmasses.

## Attribute Comparison

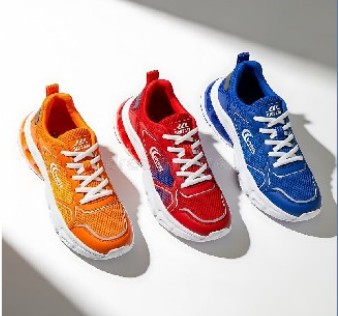

**Question:**
Comparing the three shoes from left to right, which attribute do they NOT all share in common?
A. White laces
B. White midsoles
C. Mesh upper material
D. Identical primary color
**Answer:D**

**Solution Plan**

**Step 1:** Observe the three shoes shown in the image from left to right.
**Step 2:** Compare their common features: all three have white laces (Option A), all three have chunky white midsoles (Option B), and all three feature a mesh texture on the upper parts (Option C).
**Step 3:** Identify the primary color for each shoe: the first is orange, the second is red, and the third is blue. Therefore, they do not share an identical primary color (Option D).

## Action Recognition

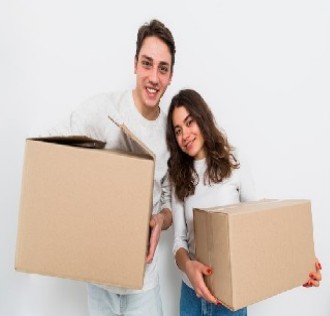

**Question:**
What primary human-object interaction is depicted in the image?
A. Two people are packing items into open cardboard boxes.
B. A man and a woman are each holding a cardboard box.
C. A couple is dragging heavy boxes across a white floor.
D. Two individuals are passing a single large box between them.
**Answer:B**

**Solution Plan**

**Step 1:** Identify the main subjects in the image, which are a man and a woman.
**Step 2:** Observe their interaction with the objects (cardboard boxes). The man is holding a large cardboard box on the left, and the woman is holding a smaller cardboard box on the right.
**Step 3:** Analyze their poses and actions. They are both standing and supporting the weight of the boxes with their hands, indicating they are holding or carrying them.
**Step 4:** Evaluate the options based on visual evidence. Option A is incorrect as the boxes are mostly closed or held in a way that doesn't show active packing. Option C is incorrect because they are holding the boxes up, not dragging them. Option D is incorrect because there are two distinct boxes, one for each person.
**Step 5:** Conclude that the correct description is that both individuals are holding a cardboard box.

# Physical Property Reasoning

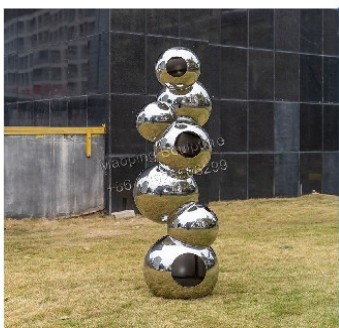

**Question:**
Based on the visual evidence in the image, what is a likely physical property of the material used to create the sculpture?
A. High specular reflectivity
B. High transparency to visible light
C. Low thermal conductivity
D. Porosity throughout the material
**Answer:A**

**Solution Plan**

**Step 1:** Observe the surface of the spherical objects in the sculpture. Notice that they show clear, detailed reflections of the surrounding grass, buildings, and sky.
**Step 2:** Identify the material properties based on these reflections. A surface that reflects light in a mirror-like way (specular reflection) is characteristic of polished metals.
**Step 3:** Evaluate the options. Option A (High specular reflectivity) directly describes this mirror-like quality. Option B is incorrect as the objects are opaque. Option C cannot be determined solely by sight, though metals usually have high conductivity. Option D is incorrect as the surfaces appear smooth and solid, not porous.
**Step 4:** Verify with auxiliary information. The title mentions 'Stainless Steel' and 'Mirror Polished', which confirms the visual observation of high specular reflectivity.

# Function Reasoning

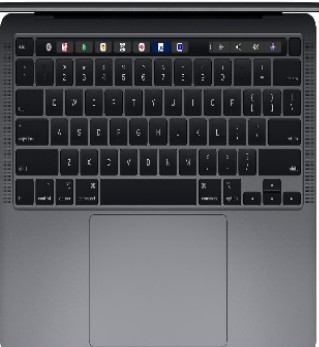

**Question:**
Based on the visual evidence in the image, what is the primary function of the main device shown?
A. Facilitating data input and user interaction via a physical interface
B. Capturing high-resolution photographs
C. Projecting video content onto a wall
D. Printing physical documents from digital files
**Answer:A**

**Solution Plan**

**Step 1:** Identify the main object in the image, which is a laptop's base assembly including a keyboard, a trackpad, and a Touch Bar.
**Step 2:** Recognize the specific components: alphanumeric keys, function keys on the Touch Bar, and a large touch-sensitive trackpad.
**Step 3:** Determine the function of these components. Keys are used for typing (data input), and the trackpad is used for cursor movement and selection (user interaction).
**Step 4:** Match this identified function with the provided options. Option B ('Facilitating data input and user interaction via a physical interface') accurately describes the purpose of a keyboard and trackpad.

# Identity Reasoning

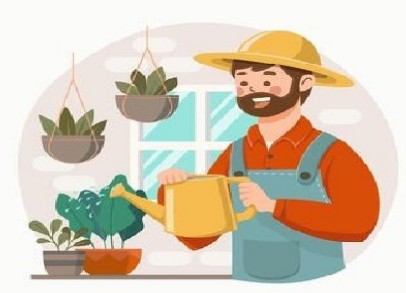

**Question:**
Based on the person's clothing and the objects they are interacting with, what is the most likely profession of the individual in the image?
A. Carpenter
B. Gardener
C. Chef
D. Painter
**Answer:B**

## Solution Plan

**Step 1**: Identify the main subject in the image, which is a man with a beard wearing a wide-brimmed yellow hat, an orange shirt, and blue overalls.
**Step 2:** Observe the environment and objects around the subject. There are multiple potted plants on a shelf and hanging from the ceiling.
**Step 3:** Analyze the subject's action. He is holding a yellow watering can and appears to be in the process of watering a plant.
**Step 4:** Synthesize these visual cues (hat for sun protection, overalls for manual work, watering can, and numerous plants) to infer the profession.
**Step 5:** Compare the inference with the provided options. A gardener is the profession most consistent with these specific tasks and tools.

# Social Relation

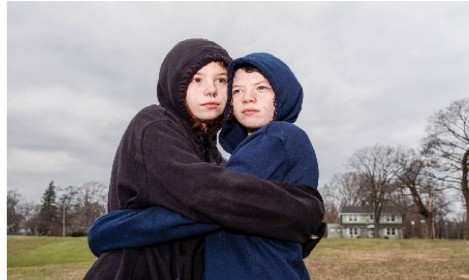

**Question:**
Based on the visual evidence, what is the most likely relationship between the two individuals in the image?
A. Siblings or peers
B. Mother and son
C. Husband and wife
D. Grandparent and grandchild
**Answer:A**

## Solution Plan

**Step 1:** Analyze the physical characteristics of both individuals in the foreground. They appear to be of similar age, likely teenagers or young adults.
**Step 2:** Observe their facial features. Both individuals have similar complexions, facial structures, and hair textures, suggesting a close biological relationship like siblings
**Step 3:** Evaluate their interaction. They are hugging tightly with solemn or protective expressions, indicating a close, supportive bond common among siblings or very close peers.
**Step 4:** Consider the auxiliary information from the title ('Do Children Have a "Right to Hug" Their Parents?'). While the title mentions parents and children, the visual evidence of the two subjects shows they are nearly the same height and age, which makes the 'Parent/Child' (Option A) or 'Grandparent/Grandchild' (Option D) relationship unlikely for these specific subjects. 'Husband and wife' (Option B) is less likely given their youthful, potentially adolescent appearance and strong family resemblance. Therefore, 'Siblings or peers' (Option C) is the most accurate description based on the image.

## Nature Relation

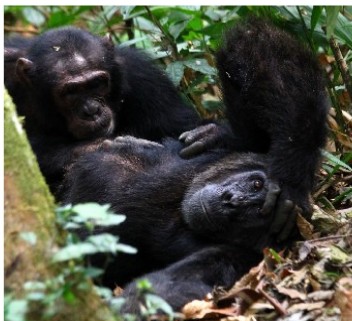

**Question:**
Observe the interaction between the two chimpanzees in the image. Which abstract biological or social relationship is primarily being demonstrated?
A. Social grooming
B. Predation
C. Interspecific competition
D. Parasitism
**Answer:A**

**Solution Plan**

**Step 1:** Identify the subjects in the image, which are two chimpanzees in a forest environment.
**Step 2:** Analyze their interaction. One chimpanzee is lying down while the other is sitting up and using its hands to pick through the fur of the first chimpanzee.
**Step 3:** Evaluate this behavior against the provided options. The act of cleaning or picking through another's fur is a classic example of social grooming (allogrooming) in primates.
**Step 4:** Cross-reference with the title 'The benefits of social grooming' and search query 'mutual grooming mammals' to confirm that this behavior is indeed social grooming, a form of cooperative social behavior rather than predation, competition, or parasitism.
**Step 5:** Conclude that option B is the correct answer based on visual evidence of the physical contact and the context of primate behavior.

## Physical Relation

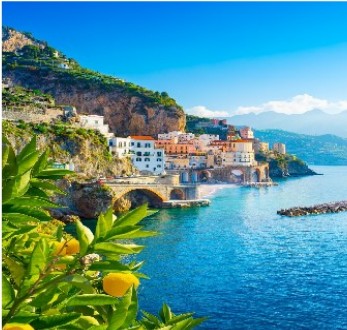

**Question:**
Based on the spatial arrangement and depth in the image, which of the following objects is physically closest to the camera lens?
A. The white buildings on the cliffside
B. The stone bridge with arches over the water
C. The mountains in the background
D. The lemons on the tree branch
**Answer:D**

**Solution Plan**

**Step 1:** Analyze the composition of the image to identify foreground, midground, and background elements.
**Step 2:** Identify the lemons on the tree branch in the bottom-left corner. Notice that they are larger in relative scale and partially overlap other elements, indicating they are in the immediate foreground.
**Step 3:** Identify the stone bridge and white buildings. These are located in the midground, further away than the lemons but closer than the mountains.
**Step 4:** Identify the hazy mountains in the distance, which represent the background.
**Step 5:** Compare the relative distances. The lemons are the closest objects to the viewer/camera.
**Step 6:** Select option C as the correct answer.

# Future Prediction

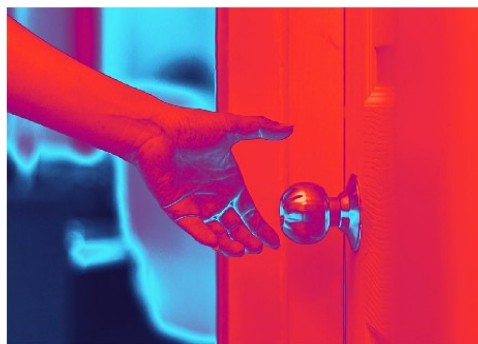

**Question:**
Based on the person's hand gesture and the proximity to the door hardware, what is the most likely immediate action that will occur?
A. The person will pull the door shut by the handle.
B. The person will knock on the wooden part of the door.
C. The person will grasp and turn the doorknob to open the door.
D. The person will use a key to unlock the deadbolt.
**Answer:C**

**Solution Plan**

**Step 1:** Identify the key objects in the image: a human hand and a round doorknob attached to a door.
**Step 2:** Observe the spatial relationship and pose: The hand is extended toward the doorknob, with the fingers beginning to curl in a reaching motion, positioned almost directly in front of the knob.
**Step 3:** Analyze the intent based on the 'reach' gesture: In a domestic setting, reaching toward a doorknob typically precedes grasping and rotating it to operate the latch.
**Step 4**: Evaluate the options: Option C aligns with the physical trajectory and target object. Option A is unlikely as the door appears mostly closed or is the frame being approached from the inside. Option B is incorrect because the hand is aimed at the hardware, not the wood. Option D is incorrect because no key is visible and the target is a knob, not a lock cylinder.

# Structuralized Image-Text Understanding

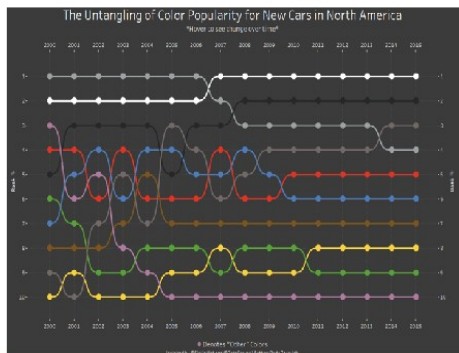

**Question:**
Based on the chart showing car color popularity in North America from 2000 to 2015, which color held the rank of #1 most popular from 2000 to 2006 before being overtaken in 2007?
A. White
B. Black
C. Silver
D. Red
**Answer:C**

**Solution Plan**

**Step 1:** Identify the chart's theme and axes. The x-axis represents years (2000-2015) and the y-axis represents rank (1-10).
**Step 2:** Locate the rank #1 line at the start of the timeline (year 2000). The line is light grey/silver.
**Step 3:** Follow this silver line from 2000 to 2006. It remains at rank #1 during this entire period.
**Step 4:** Observe the transition in 2007. The silver line drops to rank #2, while the white line rises from rank #2 to rank #1.
**Step 5:** Conclude that Silver (represented by the light grey line) was the color that held rank #1 from 2000 to 2006.

**D.2. QA Examples on the Qwen-based Version**

# Image Style

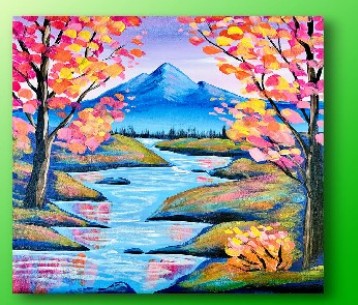

**Question:**
Based on the visual characteristics such as the visible brushstrokes, layered colors, and stylistic representation of nature, what type of image is this?
A. Photography
B. Painting
C. X-ray
D. Map
**Answer:B**

**Solution Plan**

**Step 1:** Analyze the visual textures and marks in the image. I can see clear impasto-like brushstrokes, particularly in the pink and yellow leaves of the trees and the white highlights on the water.
**Step 2:** Observe the color blending and layering. The colors are vibrant and applied in distinct patches rather than smooth photographic gradients, indicating a hand-rendered medium.
**Step 3:** Evaluate the subject matter representation. While it depicts a mountain, river, and trees, the style is expressive and stylized rather than a realistic capture of a physical scene.
**Step 4:** Compare these observations with the options. The presence of brushwork and artistic stylization identifies this as a painting.
**Step 5:** Reference the title 'Path to the Peak' and search query 'painting' as supporting evidence, confirming they align with the visual evidence of a landscape painting.

# Image Topic

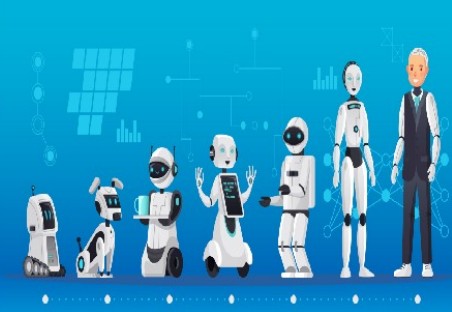

**Question:**
Which of the following best describes the subject matter of this image?
A. A group of people sitting at a table in a modern office
B. A scenic view of a futuristic city with flying vehicles
C. A close-up photograph of a single microchip
D. A collection of different robot types illustrating the evolution of AI and robotics
**Answer:D**

**Solution Plan**

**Step 1:** Observe the image content. The image shows a horizontal lineup of various mechanical and humanoid figures against a blue digital-themed background.
**Step 2:** Identify the specific objects. Starting from the left, there is a small tracked robot, a robotic dog, a service robot holding a tray, a social robot with a screen, a utility robot, a humanoid robot, and finally an android/cyborg wearing a suit.
**Step 3:** Analyze the theme. The progression from simple mechanical forms to human-like forms suggests a theme of technological evolution or variety in robotics.
**Step 4:** Compare with the options. Option B accurately describes this collection of robots, while the other options describe scenes (office, city, microchip) not present in the image.

# Image Scene

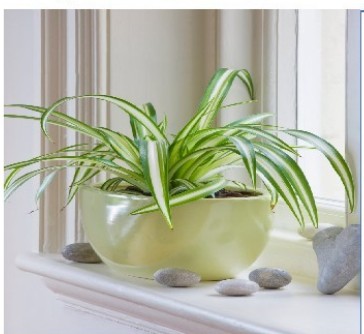

**Question:**
Based on the visual evidence in the image, what type of environment is shown?
A. Outdoors in a garden
B. Indoors on a windowsill
C. A public park
D. A deep forest
**Answer:B**

## Solution Plan

**Step 1:** Observe the main subjects and the background. The primary subject is a potted plant (spider plant) in a light green ceramic pot.
**Step 2:** Examine the immediate surface the pot is sitting on. It is a white, molded architectural feature characteristic of an interior windowsill.
**Step 3:** Look at the background structures. There are vertical frames and panels painted in a light cream color, which appear to be the casing and sash of a window.
**Step 4:** Note the presence of decorative smooth stones on the windowsill, which is common in home decor.
**Step 5:** Conclude that the setting is indoors, specifically a windowsill, rather than an outdoor or natural wild environment.
**Step 6:** Verify with the auxiliary metadata (title/search query) which mentions 'Indoor Plants', confirming the visual interpretation.

# Image Emotion

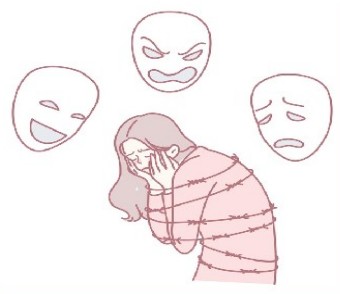

**Question:**
Which of the following best describes the overall subjective emotion conveyed by this image?
A. Joyful and celebratory
B. Peaceful and serene
C. Oppressive and distressed
D. Energetic and excited
**Answer:C**

## Solution Plan

**Step 1:** Analyze the central figure's posture and facial expression. The character is crying, has their hands covering their face, and is hunched over, indicating sadness or despair.
**Step 2:** Identify symbolic elements surrounding the figure. The person is wrapped in barbed wire, which is a common symbol for pain, restriction, or feeling trapped.
**Step 3:** Evaluate the surrounding floating masks. There are three masks with different expressions: laughing (left), angry (top), and sad (right). This suggests being judged or overwhelmed by external perceptions or internal emotional turmoil.
**Step 4:** Combine these visual cues. The combination of crying, being bound by barbed wire, and the presence of judgmental or conflicting emotional masks creates a strong feeling of being oppressed, trapped, and in distress.
**Step 5:** Select the option that matches this interpretation. 'Oppressive and distressed' accurately captures the heavy, negative emotional tone of the artwork.

# Image Quality

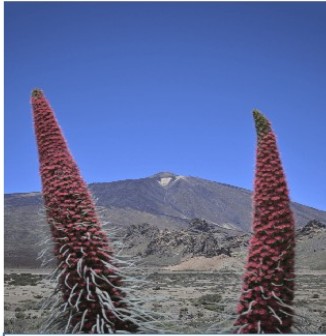

**Question:**
Which of the following best describes the overall visual characteristics of this image in terms of color and contrast?
A. A high-contrast color image showing vibrant red and blue hues.
B. A low-contrast grayscale image that is extremely dark and lacks detail.
C. A grayscale image with high contrast, clearly showing geological textures and patterns.
D. A blurred, low-resolution color image with overexposed bright spots.
**Answer:C**

**Solution Plan**

**Step 1:** Observe the color space of the image. The image is composed entirely of shades of gray, indicating it is a grayscale (monochrome) image.
**Step 2:** Assess the contrast and detail. There is a wide range of tones from deep blacks/shadows to bright whites, which creates high contrast. This contrast allows for the clear identification of fine textures like sand dunes (top right and bottom left), crater-like depressions, and rocky outcrops.
**Step 3:** Evaluate the sharpness and clarity. The image exhibits sharp edges and intricate details consistent with high-resolution satellite imagery (as hinted by the title 'HiRISE'), ruling out significant blurriness.
**Step 4:** Combine these observations to select the best descriptor. Option C correctly identifies the grayscale nature and the high-contrast detail.

# Attribute Recognition

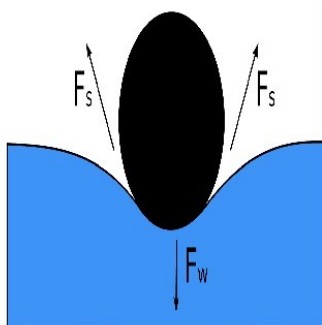

**Question:**
What is the primary shape of the object interacting with the blue surface in the image?
A. Square
B. Triangle
C. Circle
D. Hexagon
**Answer:C**

**Solution Plan**

**Step 1:** Identify the main black object positioned above the blue surface.
**Step 2:** Observe its geometric boundary and outline.
**Step 3:** Recognize that the outline is a continuous curve equidistant from a center point, which defines a circle.
**Step 4:** Confirm the choice from the provided options.

# Celebrity Recognition

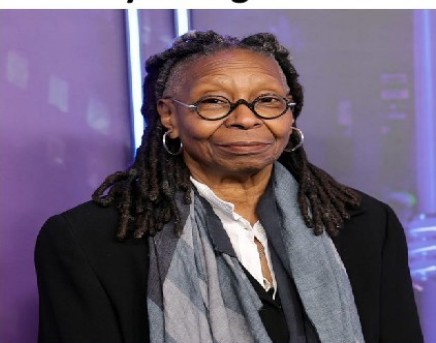

**Question:**
Who is the famous actress and television personality shown in this image?
A. Whoopi Goldberg
B. Oprah Winfrey
C. Viola Davis
D. Angela Bassett
**Answer:A**

Solution Plan

**Step 1:** Analyze the facial features, hairstyle (dreadlocks), and signature round glasses of the individual in the image.
**Step 2:** Compare these unique visual characteristics against known public figures in entertainment.
**Step 3:** Identify the individual as Whoopi Goldberg based on her iconic appearance.
**Step 4:** Verify the identification by cross-referencing with common celebrity knowledge and the provided search query context.

# Object Localization

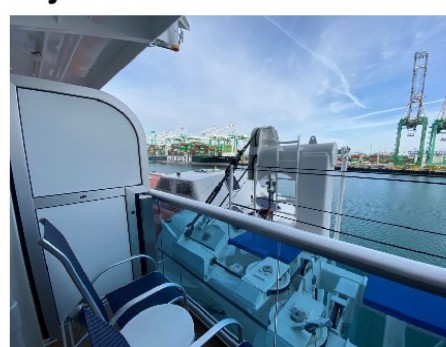

**Question:**
What is the relative position and approximate count of the green gantry cranes located on the far right side of the image, behind the ship's balcony structures?
A. Top left, 1 crane
B. Bottom right, 3 cranes
C. Bottom left, 4 cranes
D. Top right, 2 cranes
**Answer:D**

Solution Plan

**Step 1:** Identify the main setting of the image, which is a cruise ship balcony overlooking a port.
**Step 2:** Scan the background of the image to locate industrial equipment, specifically gantry cranes used for cargo.
**Step 3:** Focus on the right side of the image background to identify the green structures.
**Step 4:** Count the distinct tall green gantry crane structures visible against the sky on the far right. There are two prominent green gantry cranes visible.
**Step 5:** Determine their vertical position in the image frame. Since they are tall structures reaching into the sky area, they are in the 'top' half of the image.
**Step 6:** Combine the spatial information (top right) and the count (2) to select the correct option.

## OCR

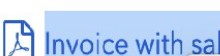

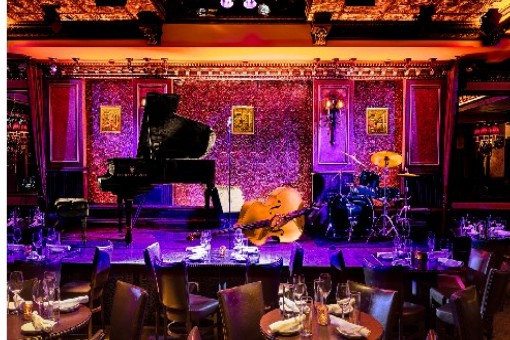

**Question:**
Based on the visual information in the image, which option in the context menu is currently highlighted or selected?
A. Open...
B. Save as...
C. View OCR text
D. Reveal file in folder
**Answer:C**

**Solution Plan**

**Step 1:** Identify the context menu appearing in the image.
**Step 2:** Scan the list of text options within the menu: 'Open...', 'Save as...', 'Reveal file in folder', 'View OCR text', and 'Copy path to clipboard'.
**Step 3:** Observe the visual styling of the text options. Note that 'View OCR text' is contained within a blue rounded rectangle, indicating it is the active/highlighted selection.
**Step 4:** Match the highlighted text to the provided multiple-choice options.

## Spatial Relationship

**Question:**
Looking at the stage from the perspective of the audience seated at the tables in the foreground, what is the position of the double bass relative to the drum kit?
A. To the right of the drum kit
B. To the left of the drum kit
C. Behind the drum kit
D. Directly in front of the drum kit
**Answer:B**

**Solution Plan**

**Step 1:** Identify the main musical instruments on the stage from the image: a grand piano, a double bass, and a drum kit.
**Step 2:** Determine the orientation/perspective. The question asks from the perspective of the audience in the foreground looking towards the stage.
**Step 3:** Locate the drum kit on the right side of the stage.
**Step 4:** Locate the double bass, which is lying on its side in the center-right area of the stage.
**Step 5:** From the audience's viewpoint, compare the horizontal positions. The double bass is positioned to the left of the drum kit.

# Attribute Comparison

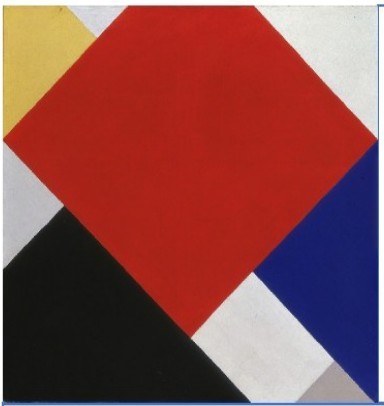

**Question:**
Comparing the colored geometric shapes in the image, which shape occupies the largest area of the composition?
A. The yellow triangle in the top-left corner
B. The blue rectangle on the right side
C. The red square rotated in the center
D. The black triangle in the bottom-left corner
**Answer:C**

## Solution Plan

**Step 1**: Identify all major colored geometric shapes in the image: a yellow triangle (top-left), a large red square rotated 45 degrees (center), a blue shape (right), and a black shape (bottom-left).
**Step 2:** Visually compare the relative size and area occupied by each shape.
**Step 3:** Observe that the red square is the central focus and covers significantly more surface area than any of the peripheral shapes (yellow, blue, black, or white sections).
**Step 4:** Conclude that the red central square is the largest shape.

# Action Recognition

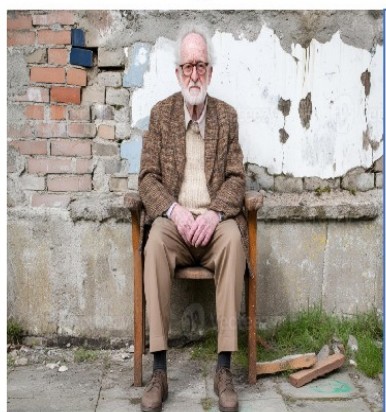

**Question:**
Which of the following best describes the human pose and interaction depicted in the image?
A. An elderly man is standing upright while leaning against a damaged brick wall.
B. An elderly man is sitting on a wooden chair with his hands clasped in his lap.
C. An elderly man is sitting on a bench and actively reading a newspaper.
D. An elderly man is crouching on the ground to inspect some broken wooden planks.
**Answer:B**

## Solution Plan

**Step 1:** Identify the main subject in the image, which is an elderly man with white hair and glasses.
**Step 2:** Analyze the person's pose. The man is in a seated position, as his knees are bent and his torso is supported.
**Step 3:** Identify the object he is interacting with. He is sitting on a wooden armchair with visible armrests.
**Step 4:** Examine the hand position. His hands are resting on his lap, clasped together.
**Step 5:** Compare these visual observations with the provided options to select the most accurate description.

# Physical Property Reasoning

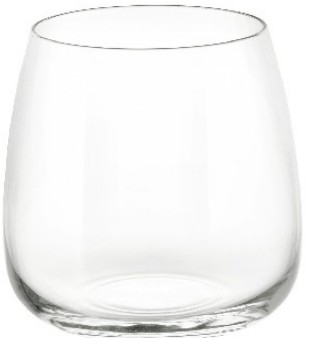

**Question:**
Based on the image, what is a primary physical property of the material used to make the object shown?
A. High electrical conductivity
B. High transparency to visible light
C. Malleability at room temperature
D. Opaque and metallic luster
**Answer:B**

**Solution Plan**

**Step 1:** Identify the object in the image as a drinking vessel (glass).
**Step 2:** Observe the visual characteristics of the material. The background is clearly visible through the object, and there are highlights reflecting light off a smooth, colorless surface.
**Step 3:** Determine the material type based on these visual cues (glass).
**Step 4:** Evaluate the physical properties of glass. It is known for its transparency, brittleness, and electrical insulating properties.
**Step 5:** Match the observed property (transparency) to the options provided. Option B correctly identifies 'High transparency to visible light'.

# Function Reasoning

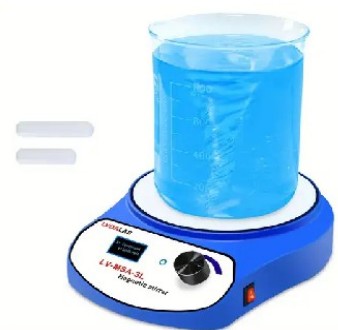

**Question:**
Based on the image, what is the primary function of the blue device shown?
A. Heating liquids to a boil
B. Measuring the weight of the beaker
C. Agitating or mixing a liquid solution
D. Filtering solids out of a mixture
**Answer:C**

**Solution Plan**

**Step 1:** Identify the main object in the image, which is a blue base with a beaker of blue liquid on top.
**Step 2:** Observe the text on the front panel of the device, which explicitly states 'Magnetic stirrer' and 'LV-MSA-3L'.
**Step 3:** Look for visual cues of the device's operation, such as the vortex visible in the blue liquid, indicating motion.
**Step 4:** Identify the two white capsule-shaped objects to the left, which are magnetic stir bars used in conjunction with a magnetic stirrer to mix liquids.
**Step 5:** Conclude that the primary function of a 'Magnetic stirrer' as shown is to agitate or mix liquid solutions.

# Identity Reasoning

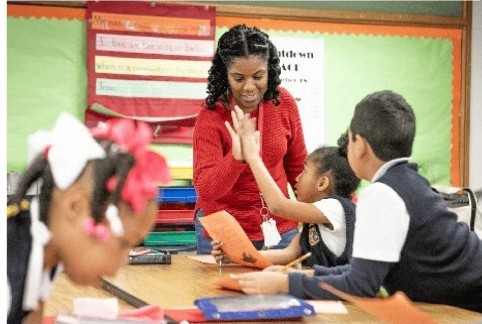

**Question:**
Based on the visual cues in the image, such as the setting, the interaction with children, and the items on the wall, what is the most likely occupation of the woman in the red sweater?
A. Teacher
B. Doctor
C. Chef
D. Construction Worker
**Answer:A**

**Solution Plan**

**Step 1:** Identify the main subjects in the image, which include a woman in a red sweater and several children.
**Step 2:** Observe the setting and background. There are educational posters on the wall (e.g., 'I live in the city of Dallas', 'Texas'), a green bulletin board, and a classroom-like desk setup.
**Step 3:** Analyze the woman's activity and appearance. She is giving a high-five to a student, holding a piece of paper, and wearing an ID badge around her neck, which is common for staff in educational institutions.
**Step 4:** Combine these observations (classroom environment, educational materials, mentoring/encouraging behavior) to infer that she is a teacher.
**Step 5:** Compare this inference with the provided options and select the best fit.

# Social Relation

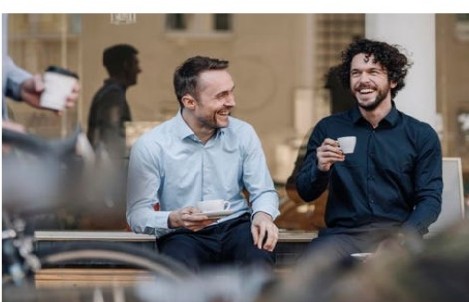

**Question:**
Based on the visual evidence, which of the following best describes the social relationship between the two men seated and holding coffee cups?
A. Competitors in a professional sport
B. Close relatives like father and son
C. A doctor and a patient in a consultation
D. Colleagues or friends sharing a casual break
**Answer:D**

**Solution Plan**

**Step 1:** Identify the main subjects in the image, which are two men sitting next to each other.
**Step 2:** Observe their attire and the setting. Both men are wearing collared shirts (business casual), and they are in a public or semi-public space, likely an outdoor seating area or a cafe.
**Step 3:** Analyze their actions and expressions. Both men are holding small coffee cups (espresso cups) and are laughing/smiling while looking at each other, indicating a positive and friendly social interaction.
**Step 4:** Integrate the visual evidence with the provided metadata. The search query 'colleagues' and the title 'Employees Want More Friends At Work' support the interpretation that these individuals are coworkers or friends.
**Step 5:** Evaluate the options. 'Competitors' (A) is unlikely given the relaxed setting. 'Father and son' (B) is less likely as they appear to be of similar adult age groups. 'Doctor and patient' (D) is incorrect based on the casual environment and identical activities (drinking coffee). 'Colleagues or friends' (C) fits the visual context of a shared social break.

# Nature Relation

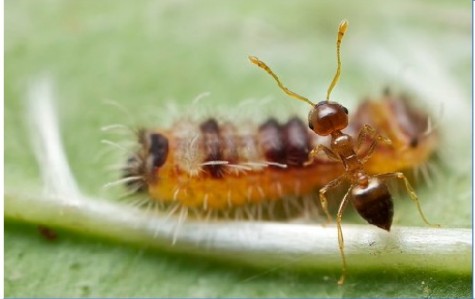

**Question:**
Based on the visual evidence in the image and the context of biological interactions between these specific types of insects, what is the most likely ecological relationship being depicted?
A. Predation
B. Mutualism
C. Competition
D. Parasitism
**Answer:B**

**Solution Plan**

**Step 1:** Identify the two main organisms in the image. There is a reddish-brown ant in the foreground and a fuzzy, orange and brown caterpillar in the background.
**Step 2:** Observe the interaction. The ant is standing near or on the caterpillar without attacking it, and the caterpillar shows no signs of defensive posturing or injury. They appear to be coexisting in close proximity.
**Step 3:** Reference the auxiliary information. The title and search query both mention 'mutualism'. In biology, many species of ants and lycaenid caterpillars have a mutualistic relationship where the caterpillar provides sugary secretions (honeydew) to the ant, and the ant provides protection from predators.
**Step 4:** Verify the visual cues against the options. Predation (A) is unlikely as neither is attacking the other. Competition (C) usually involves resources and isn't typically shown by direct physical contact like this. Parasitism (D) involves one living inside or strictly off the other's tissue, whereas this looks like a surface-level interaction common in mutualistic ant-caterpillar relations.
**Step 5:** Conclude that mutualism is the most accurate description of this ecological relationship.

# Physical Relation

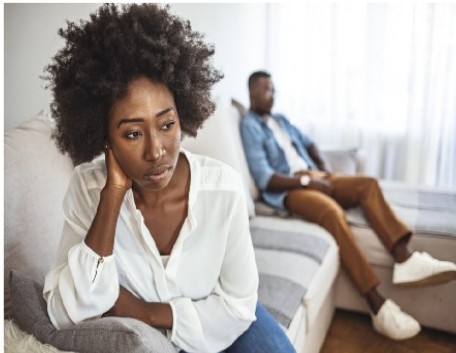

**Question:**
In the image, where is the woman positioned relative to the man sitting on the sofa?
A. She is in the foreground, closer to the camera than the man.
B. She is in the background, further from the camera than the man.
C. She is sitting directly behind the man.
D. They are sitting side-by-side at the same distance from the camera.
**Answer:A**

**Solution Plan**

**Step 1:** Identify the two main subjects in the image: a woman in the foreground and a man in the background.
**Step 2:** Observe the depth cues. The woman is sharply in focus, larger in the frame, and her body overlaps the space in front of the sofa section where the man is sitting.
**Step 3:** Observe the man, who is out of focus (bokeh effect) and appears smaller, indicating he is at a greater distance from the camera.
**Step 4:** Conclude that the woman is in the foreground and closer to the camera than the man.
**Step 5:** Match this conclusion to the corresponding option.

## Future Prediction

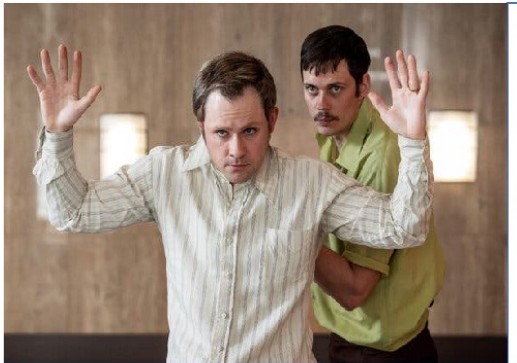

**Question:**
Based on the scene where the man in the striped shirt has his hands raised and looks intensely forward while another man stands closely behind him, what is the most likely immediate event to occur?
A. The man in front will be searched or apprehended.
B. They will start dancing together.
C. They will both sit down to have a meal.
D. The man in the back will help the man in front put on a jacket.
**Answer:A**

**Solution Plan**

**Step 1:** Analyze the poses of the individuals. The man in the foreground (striped shirt) has his hands raised in a 'surrender' or 'hands up' position, which is a common physical indicator of being confronted or apprehended.
**Step 2:** Observe the man in the background (green shirt). He is positioned closely behind the first man with a serious, focused expression, suggesting a state of tension or control.
**Step 3:** Evaluate the overall context and facial expressions. The intense, wary looks and the specific 'hands up' gesture strongly suggest a law enforcement, criminal, or high-stakes confrontation scenario.
**Step 4:** Predict the future action. Given the posture of surrender and the proximity of the person behind, the most logical next step is a search, a pat-down, or an arrest/apprehension.

# Structuralized Image-Text Understanding

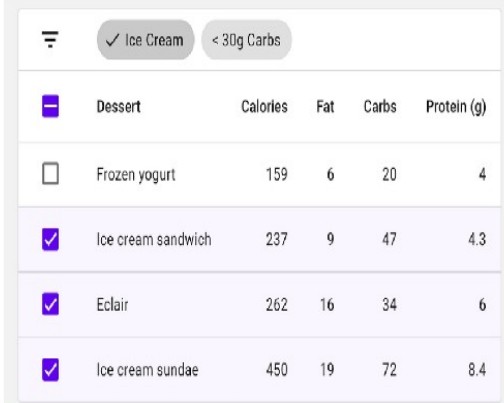

**Question:**
Based on the data table provided in the image, which dessert has the highest amount of fat and what is that value?
A. Frozen yogurt, 6
B. Ice cream sandwich, 9
C. Eclair, 16
D. Ice cream sundae, 19
**Answer:D**

## Solution Plan

**Step 1:** Locate the 'Fat' column in the data table.
**Step 2:** Read the fat values for each listed dessert: Frozen yogurt (6), Ice cream sandwich (9), Eclair (16), and Ice cream sundae (19).
**Step 3:** Compare these numerical values to find the maximum.
**Step 4:** Identify that 19 is the highest value, which corresponds to the 'Ice cream sundae'.

