# OpenReview forum: "MMBench-Live: A Continuously Evolving Benchmark for Multimodal Models"
_ICML.cc/2026/Conference — ICML 2026 regular_

### Official Review · Reviewer_D7x3 · 2026-03-11

**Soundness:** 3
**Presentation:** 2
**Significance:** 3
**Originality:** 3
**Overall Recommendation:** 3
**Confidence:** 4

**Summary:**

The authors introduce MMBench-Live, a dynamically evolving multimodal benchmark driven by a multi-agent framework. They propose a distribution-consistent update strategy that preserves the core data characteristics of the original benchmark while incorporating new data. The authors also evaluate a series of open-source vision–language models on the benchmark.

**Compliance With Llm Reviewing Policy:**

Affirmed.

**Final Justification:**

The authors addressed some of my questions. However, due to the uncertainty of the new data contamination test results, the limited amount of data on MME(7 samples), and the lack of reported model evaluation results on MME, I will maintain my original score for the sake of rigor. Considering that the authors included detailed experiments and analysis in the rebuttal, I will increase the corresponding score for soundness.

**Key Questions For Authors:**

1. In Section 3.1, the paper states that the Benchmark Summary is constructed based on MMBench, while the upper-right part of Figure 1 shows that it is generated by the Benchmark Summary Agent. Is this component manually constructed or automatically generated by the agent?

2. What prompts are used by the agent during the generation process?

3. What are the proportions of data generated by the three Acquisition Executors, respectively?

4. Is it possible to construct multiple versions of the dynamic benchmark? If so, what is the correlation of model performance across different dynamically generated benchmark versions?

5. The authors claim that the proposed framework is generalizable. How well does it generalize to other base datasets?

**Limitations:**

yes

**Strengths And Weaknesses:**

Strengths:

1. The authors introduce a dynamically evolving multimodal benchmark driven by a multi-agent framework and propose an update strategy to ensure that the generated data remain distributionally consistent with the original dataset.

2. The authors evaluate multiple models and conduct analytical experiments, including an investigation of potential data contamination issues.

3. The paper provides illustrative examples that help facilitate understanding of the proposed approach.


Weaknesses:

1. The benchmark defines five semantic perspectives inspired by **MMBench**, visual content, visual style, OCR dependency, spatial relations, and external knowledge dependency. While these dimensions are closely aligned with MMBench, they are manually specified. How do these choices influence the final data generation process? Since the authors claim that the proposed framework has strong generalizability. If the framework were applied to a different base dataset, how should these manually designed components be adapted or redesigned?

2. The implementation details are not clearly described in several parts of the paper. For example, in **Executable Reasoning Verification**, what exactly does each *tool invocation operating on the image* refer to? Does it involve calling an MLLM to produce answers? Additionally, what model is used for the textual verifier?

3. The overall construction process relies heavily on closed-source models such as **GPT-5** and **Gemini**. This may introduce potential model biases. Moreover, since the questions are generated by these models, evaluating the same models on the generated benchmark may introduce preference bias. It would be helpful to report the performance of the generation models themselves (e.g., GPT-5, Qwen-VL) on the benchmark.

4. The overall pipeline appears to retrieve images from external sources that are pattern similar to those in the original dataset and then automatically generate questions using large models. This construction process is relatively pipeline-based, and the diversity of images may still be constrained by the data sources.

5. There may be issues with the p-value calculation reported in Table 3. **PaCoST** states that *“p < 0.05 represents statistically significant and probably contaminated, while p ≥ 0.05 represents uncontaminated.”* The experiments related to data contamination may require further clarification and explanation.

6. VLB[1] has also conducted experiments on MMBench. How does MMBench-Live compare with VLB in terms of distribution consistency with the original MMBench dataset and robustness to data contamination?

[1]DYNAMIC MULTIMODAL EVALUATION WITH FLEXIBLE COMPLEXITY BY VISION-LANGUAGE BOOTSTRAPPING (ICLR2025)

---

> ### Author Rebuttal · Authors · 2026-03-31
>
> We appreciate the reviewer’s careful reading and constructive feedback. Below, we provide clarifications and additional explanations to address these points.
>
> 1. Implementation Details (to w1, w2, q1, q2, q3)
>
> Thanks for this suggestion. We clarify the main implementation details as follows. Semantic perspectives guide both query construction and filtering: they identify task-relevant visual attributes for search generation and then act as filtering criteria for retrieved images. The resulting mismatch feedback is used to refine queries in later rounds. Executable Reasoning Verification relies on specialized vision modules: VisionReasoner for detection and segmentation, Depth Anything V2 for depth estimation, a fine-tuned LLaVA for attribute recognition, Qwen2.5-VL plus GOT-OCR-2-HF for OCR, and Qwen3 as the textual verifier that aggregates intermediate outputs and performs final reasoning. Benchmark Summary is produced by the Benchmark Summary Agent, which fills a manually defined structured template using information from MMBench. Agent prompts are omitted here due to space limits, but will be released in the revised appendix and codebase. Acquisition executors are used in a ratio of 1:15:4 for source-specific, keyword-guided, and open-domain retrieval.
>
> 2. Comparison with VLB (to w6)
>
> Thank you for the reviewer’s suggestion to compare our method with VLB. The distribution consistency results for VLB are as follows.
>
> | Method | FID | KL(A∥B) | KL(B∥A) |
> | --- | --- | --- | --- |
> | Add | 0.080 | 51.62 | 29.29 |
> | Remove | 0.026 | 17.17 | 10.51 |
> | Outpainting | 0.088 | 50.22 | 39.65 |
>
> VLB uses image editing, preserving low-level statistics and thus yielding lower FID/KL. In contrast, MMBench-Live collects new real-world images, resulting in slightly higher visual distance but greater temporal novelty and diversity. We also provide additional contamination results.
>
> | Method | Mean | Std | p | Cohen’s d |
> | --- | --- | --- | --- | --- |
> | Add | 0.042 | 0.187 | 6.1e-14 | 0.224 |
> | Remove | 0.038 | 0.200 | 3.7e-12 | 0.191 |
> | Outpainting | 0.042 | 0.205 | 1.1e-13 | 0.205 |
> | MMBench-Live | **0.006** | **0.102** | 2.9e-6 | **0.059** |
>
> Compared with VLB variants, MMBench-Live shows much smaller confidence shifts and effect sizes. Together with temporal isolation, this suggests that our live-update paradigm provides a more robust way to reduce contamination risk than editing.
>
> 3. Clarification on Contamination Evaluation (to w5)
>
> Thank you for pointing this out. We note that PaCoST was originally developed for LLMs, so relying solely on p-values may be insufficient in multimodal settings. To examine this further, we conducted additional VLB-style perturbation experiments, where all variants also yield p<0.05, suggesting limited discriminative power of this criterion here. In addition, PaCoST paraphrasing often removes visual grounding cues, which can reduce image dependency and lower confidence for some samples. We will add a clarification of this point in the revised version.
>
> 4. Generalizability of the Framework (w1, q5)
>
> Due to the rebuttal space limit, we refer the reviewer to our response to **Reviewer YaYB** under “Generalizability of the Framework” for this point.
>
> 5. Potential Model Biases (to w3)
>
> Thank you for raising this important concern. We additionally test GPT-5-mini on QA pairs generated by Gemini-3-flash-preview. GPT achieves 95.19%, while Gemini achieves 91.54%, suggesting that the benchmark is not simply biased toward its question generator. Although these scores are relatively high, they are not saturated. Together with the clear performance separation among open-source models, this suggests that any preference bias is limited and does not dominate the evaluation outcome.
>
> 6. Data Sources Diversity (to w4)
>
> Thank you for the valuable feedback. Our pipeline retrieves images aligned with the original benchmark patterns, but it is not tied to a fixed source distribution. Since MMBench does not provide explicit source links, we are not replicating predefined sources. Instead, we use three complementary acquisition modes: source-specific, keyword-guided, and open-domain retrieval, introducing more diverse and temporally updated visual content.
>
> 7. On Multi-Version Benchmark Construction and Correlation（to q4）
>
> Our framework supports constructing multiple versions by controlling the data acquisition window. Cross-version consistency can then be measured by Pearson or Spearman correlation at the model level, enabled by our distribution-consistent updating strategy that reduces capability shifts across versions. A more principled option is to use IRT to estimate question difficulty and latent model ability for each version, and then compare the resulting ability estimates across versions. However, this usually requires proper scale linking or anchor design, since fully independent versions may not share a directly comparable latent scale. We will explore this direction in future work.

---

> > ### Author Rebuttal · Reviewer_D7x3 · 2026-04-03
> >
> > Thanks for the authors’ response. Some of my concerns have been addressed; however, I still have a few remaining questions.
> >
> > 1. Throughout the pipeline, the authors employ a large number of both open- and closed-source models, including VisionReasoner, Depth Anything V2, LLaVA, Qwen2.5-VL, GOT-OCR-2-HF, and Qwen3. Could hallucinations arise at individual stages, and could errors accumulate across the overall pipeline? The authors are encouraged to provide a more fine-grained, step-by-step error analysis.
> >
> > 2. Is the data contamination analysis for VLB still conducted using PaCoST? If so, its reported p-value remains below 0.05, which suggests that the current contamination analysis is still not sufficiently rigorous.
> >
> > 3. Since most of the data are obtained via keyword-guided retrieval, and this part of the dataset is collected through the Google Image API, the retrieved data may change over time. For some images, however, the updates may be very limited over long periods, which could weaken the dynamic nature of the dataset. Under such circumstances, can the framework still continuously generate dynamic test data?
> >
> > 4. Given that the original contamination analysis based on PaCoST is not sufficiently rigorous, beyond adding further explanation, could the authors supplement it with additional contamination analysis experiments?
> >
> > 5. Regarding generalization, could the authors provide results on samples drawn from other benchmarks to further validate the method’s generalizability?
> >
> > 6. Regarding model bias, it would also be helpful to report or supplement results on QA pairs generated by GPT-5-mini, comparing the performance of GPT and Gemini on those samples, and further contrasting these results with their performance on QA pairs generated by Gemini.
> >
> > 7. For different versions of the benchmark, a dynamic evaluation framework should indeed provide dynamic updates to mitigate data contamination. At the same time, it should also yield convergent evaluation conclusions across different dynamic settings and test batches; otherwise, if each dynamic evaluation under different versions or dynamic settings or just generated in different times, leads to inconsistent conclusions, it becomes difficult to derive stable and reliable assessment results.

---

> > > ### Author Response · Authors · 2026-04-08
> > >
> > > To q1: Step-by-Step Error Analysis
> > >
> > > Thank you for this important comment. We agree that hallucinations or incorrect reasoning can arise at different stages of our multi-stage pipeline, and such errors may accumulate. In practice, specialized tools help reduce perception-stage errors, while failures are more likely to occur in the final aggregation-and-reasoning stage, where Qwen3 integrates tool outputs to derive the answer.
> > >
> > > For example, in the question “Comparing the toy cars in the image, which statement regarding their colors and positions is correct?”, one option states: “There is a yellow car and an orange car in the same diagonal row starting from the left.” In one failure case, the perception module correctly identified the toy cars and their rotated grid layout, and the upstream tools also correctly returned the relevant color attributes and bounding boxes. However, the final reasoning stage still incorrectly concluded that the two cars lie on the same diagonal, whereas under the rotated grid structure, the up-right direction should instead be interpreted as a row. This suggests that the main error source in this case lies in the final relational reasoning stage rather than in perception or attribute extraction. We will add a more fine-grained stage-wise discussion of such errors in the revision.
> > >
> > > To q2、q4: Contamination Analysis
> > >
> > > Yes, the contamination analysis for VLB is conducted using the same PaCoST-based protocol. We agree that PaCoST should be viewed as a proxy-based analysis rather than a definitive contamination test. Therefore, while its results can provide suggestive evidence, they are not sufficient on their own to support strong conclusions about multimodal contamination. We will clarify this limitation more explicitly in the revision. Beyond PaCoST, we plan to complement the current analysis with additional contamination detection methods and a broader multi-perspective assessment, rather than relying on a single proxy alone.
> > >
> > > To q3: Dynamic Data Collection
> > >
> > > Thank you for this important question. Our framework does not rely on a single data source or a fixed set of keyword queries to sustain benchmark updates. Although keyword-guided retrieval is the dominant mode in the current implementation, it is only one component of the overall acquisition pipeline. More importantly, continuous evolution is maintained at the benchmark level, rather than requiring every individual query to remain productive over time. Even if some queries become saturated or some images update only slowly, the framework can still collect new task-relevant data by refining queries, expanding the keyword set, and targeting newly emerging entities or events. Therefore, limited updates in some individual queries do not prevent the benchmark from continuously generating newer test data.
> > >
> > > To q5: Generalization
> > >
> > > We have conducted a preliminary transfer experiment on MME. In this setting, the only required manual step is lightweight dataset preprocessing. Although this pilot experiment currently produces a small number of transferred samples, it provides preliminary evidence that the framework is not tightly coupled to a single benchmark and can be adapted to other benchmarks with minimal manual effort. We provide a small set of anonymous example images and QA pairs at https://anonymous.4open.science/r/examples-D696.
> > >
> > > To q6: Model Bias
> > >
> > > We appreciate this suggestion. We additionally evaluate both models on QA pairs generated by GPT-5-mini. On these samples, GPT-5-mini achieves 97.9%, while Gemini achieves 96.0%. Together with the results on Gemini-generated QA pairs, this indicates that the benchmark is not simply biased toward a particular question generator, although we do not rule out the possibility of limited residual generator-specific effects.
> > >
> > > To q7: Cross-Version Stability of Dynamic Evaluation
> > >
> > > We appreciate this suggestion. To examine this, we construct MMBench-Live-v2, a second dynamic version of MMBench-Live. Specifically, we collect a new set of 5K images using the same framework and use perceptual hashing to ensure that the images do not overlap with those in MMBench-Live-v1.
> > >
> > > The model results on MMBench-Live-v2 are shown below:
> > >
> > > | Model | acc |
> > > | --- | --- |
> > > | deepseek-vl | 84.19% |
> > > | instructblip-7b | 75.58% |
> > > | llava-7b | 76.84% |
> > > | Qwen3-VL-8B | 86.58% |
> > > | mplug_owl2 | 78.75% |
> > > | Qwen2.5-VL-7B | 85.97% |
> > >
> > > We further compute the cross-version correlation with the original MMBench:
> > >
> > > | **Granularity** | **Pearson r** | **Pearson p** | **Spearman ρ** | Spearman p |
> > > | --- | --- | --- | --- | --- |
> > > | Task-Model Level | 0.6039 | 2.835e-13 | 0.5568 | 3.995e-11 |
> > > | Model-Averaged Level | 0.9534 | 0.003203 | 0.9429 | 0.004805 |
> > >
> > > These results suggest high consistency at the task-model level and model-averaged level. This supports our claim that the framework can introduce temporally new test data while preserving convergent evaluation conclusions at the overall model-comparison level.

---

### Official Review · Reviewer_YaYB · 2026-03-12

**Soundness:** 3
**Presentation:** 3
**Significance:** 3
**Originality:** 3
**Overall Recommendation:** 4
**Confidence:** 3

**Summary:**

This paper presents MMBench-Live, a continuously evolving multimodal benchmark designed to address the limitations of static VLM evaluation, especially data contamination, temporal staleness, and high manual construction cost. The framework uses a multi-agent pipeline that converts the original benchmark into structured task descriptions, acquires new task-aligned data, and generates QA pairs with executable verification. Based on this pipeline, the authors build an updated version of MMBench and evaluate several open-source VLMs, analyzing benchmark quality, cross-version consistency, and contamination effects. The results suggest that the proposed benchmark can maintain relatively stable evaluation behavior while reducing contamination and construction cost.

**Compliance With Llm Reviewing Policy:**

Affirmed.

**Final Justification:**

The authors have addressed some of my concerns.

**Key Questions For Authors:**

1. The main novelty is moderate. The work is more of a benchmark construction and maintenance framework than a strong methodological contribution on the modeling side.

2. The whole pipeline depends heavily on strong proprietary or frontier models for planning, judging, QA generation, and verification, which raises concerns about reproducibility and system bias.

3. The contamination analysis is suggestive but not fully conclusive. A smaller confidence shift on the updated benchmark does not by itself fully establish lower contamination.

4. The evaluation is still centered on MMBench, so the claim that the framework is broadly general and extensible to other multimodal benchmarks is not yet fully validated.

5. The writing needs improvement. There are noticeable grammar problems, awkward phrasing, and presentation issues in multiple sections.

**Limitations:**

see above

**Strengths And Weaknesses:**

1. The paper studies an important problem. Static multimodal benchmarks do become less reliable over time due to contamination and staleness.

2. The overall pipeline is clear and fairly complete, covering benchmark summarization, task-aware data acquisition, QA generation, and executable verification.

3. The idea of preserving cross-version consistency while introducing temporally newer samples is meaningful and practically useful.

4. The paper includes several useful analyses, including QA correctness, construction cost, contamination testing, and ablations on visual patterns and feedback.

5. The reported construction efficiency is impressive, and the framework appears much cheaper and faster than a manual benchmark refresh.

---

> ### Author Rebuttal · Authors · 2026-03-31
>
> We appreciate the reviewer’s careful reading and constructive feedback. The comments on the contribution, reproducibility, contamination analysis, generalizability, and presentation are very helpful. Below, we provide clarifications on these points.
>
> 1. Clarification on the Contribution (to q1)
>
> We appreciate the comment. Our work addresses the problem of benchmark evolution and maintenance for multimodal evaluation. Within this scope, we view the contribution as methodological and novel. Existing static benchmarks are increasingly limited by data contamination, temporal staleness, and high update cost, while recent dynamic evaluation approaches often suffer from semantic drift, distribution mismatch, or reliance on fixed pipelines. In contrast, our framework formulates benchmark updating as a task-guided, distribution-consistent construction process. Its novelty lies not simply in automating benchmark refresh, but in enabling temporal updates while preserving the core characteristics of the original benchmark through structured specification, task-related visual patterns, and feedback-driven filtering. We therefore view the work as contributing a general methodology for multimodal benchmark construction and maintenance.
>
> 2. Reproducibility and Model Dependence (to q2)
>
> Thank you for this important comment. Our framework is designed at the functional level rather than around any specific proprietary system. The key components—task specification, data acquisition, judging, generation, and verification—are defined by their roles in the pipeline, which allows them to be instantiated with different models. To validate this, we conduct an ablation study in which the key components in task specification and data acquisition are replaced with open-source Qwen-based models. As shown in Table 5, the resulting pipeline still preserves clear performance differentiation across evaluated models and maintains cross-version consistency. These results suggest that the effectiveness of the framework comes from the overall pipeline design, rather than dependence on a particular frontier model.
>
> 3. Clarification on the Contamination Analysis (to q3)
>
> Thank you for the comment. Accurately assessing data contamination remains challenging, especially in multimodal settings, due to the limited transparency of large-scale pretraining corpora. In this context, our analysis should be understood as a proxy based on memorization effects rather than a direct measurement of contamination itself. The reduced confidence shift, together with the smaller effect size and statistically significant trends, provides empirical evidence that MMBench-Live is less affected by contamination than the original MMBench. In addition, our pipeline already reduces contamination risk at the data-collection stage by restricting acquisition to recent samples. The confidence-shift analysis is intended as an additional validation signal on top of this design, and together these factors provide coherent evidence that the updated benchmark is less vulnerable to contamination.
>
> 4. Generalizability of the Framework (to q4)
>
> Our framework is designed as a task-centric dataset construction process, and its generalizability mainly comes from three components: a structured benchmark template, task-related visual patterns, and flexible data acquisition strategies. The template provides a unified schema for decomposing benchmarks into sub-tasks and supporting data acquisition and instance generation. The task-related visual patterns are generic descriptors rather than dataset-specific rules: although inspired by MMBench, the perspectives themselves are broadly applicable to most vision tasks, and their concrete content is automatically derived by the agent from the target benchmark. In addition, the acquisition module includes three retrieval modes, making the framework adaptable to different image-text benchmarks. At the current stage, however, it is primarily designed for image-text benchmarks; extending it to other modalities would require adding corresponding tools and adapting the current patterns to the new modality.
>
> 5. Writing and Presentation (to q5)
>
> Thank you for the comment. We will carefully revise the paper to improve grammar, phrasing, and overall presentation. In the revision, we will proofread the manuscript thoroughly, correct awkward or unclear expressions, and improve readability across sections. We will also further polish the organization and presentation to make the paper clearer and more concise.

---

> > ### Author Rebuttal · Reviewer_YaYB · 2026-04-01
> >
> > After reading the rebuttal, I still have some questions:
> > 1. How to achieve Continuously Evolving as the world always changes with time changes?
> > 2. Is there any method to measure the data Contamination and give some qualitative results?
> > 3. How many human efforts are needed in the benchmark construction? Could others repeat this pipeline easily?

---

> > > ### Author Response · Authors · 2026-04-08
> > >
> > > To q1: How Continuous Evolution is Achieved
> > >
> > > Our framework achieves continuous evolution by treating benchmark updating as a repeatable task-guided data construction process rather than a one-time dataset release. Specifically, the benchmark can be refreshed by repeatedly: (1) using the benchmark summary and task descriptions to specify the target data scope, (2) retrieving newly available data within a controlled time window, (3) filtering the data based on task-related visual patterns and model feedback, and (4) automatically generating and verifying new QA pairs. In this way, as the world changes over time, the framework can continuously incorporate newer time-sensitive data while preserving the original task definition, evaluation objective, and construction protocol. Therefore, “continuously evolving” means systematic regeneration under a fixed evaluation framework, rather than arbitrary changes to the benchmark.
> > >
> > > To q2: Data Contamination
> > >
> > > We agree that PaCoST should be viewed as a proxy-based analysis rather than a definitive contamination test. Therefore, while its results can provide suggestive evidence, they are not sufficient on their own to support strong conclusions about multimodal contamination or to provide reliable qualitative evidence in this setting. At the current stage, we do not yet have sufficiently robust direct evidence to report strong qualitative contamination results. We will clarify this limitation more explicitly in the revision. Beyond PaCoST, we plan to complement the current analysis with additional contamination detection methods, such as image-level retrieval of benchmark images and their near-duplicate variants in public pretraining-scale datasets, together with a broader multi-perspective assessment rather than relying on a single proxy alone.
> > >
> > > To q3: Human Effort and Reproducibility
> > >
> > > Human effort in our benchmark construction pipeline is minimal. In our preliminary transfer experiment on MME, which produced only a small number of samples, the only required manual operation was lightweight dataset preprocessing, i.e., converting the original MME data into the unified JSON format required by the pipeline. After this initialization step, the remaining stages are handled automatically by the framework. This design also makes the pipeline easy to reproduce: once a benchmark is converted into the same input format, the subsequent construction procedure can be reused directly. We provide a small set of anonymous example images and QA pairs at https://anonymous.4open.science/r/examples-D696
> > > , and we will release the implementation code and constructed data to facilitate reproduction and extension by other researchers.

---

### Official Review · Reviewer_a7FG · 2026-03-12

**Soundness:** 3
**Presentation:** 3
**Significance:** 3
**Originality:** 3
**Overall Recommendation:** 4
**Confidence:** 3

**Summary:**

This paper proposes MMBench-Live, a continuously updated multimodal benchmark built from the original MMBench through an automated pipeline consisting of structured benchmark summarization, task-aware data acquisition, and executable QA generation with verification. The benchmark can be refreshed over time without human annotation while preserving capability coverage, distributional alignment, and cross-version comparability. Empirically, the paper reports 5.9K new QA pairs, a manual QA correctness estimate of 96.06%, strong model ranking consistency between MMBench and MMBench-Live, and a reduced contamination signal under PaCoST relative to the original benchmark.

**Compliance With Llm Reviewing Policy:**

Affirmed.

**Key Questions For Authors:**

Please see Weaknesses

**Limitations:**

Please see Weaknesses

**Strengths And Weaknesses:**

### Strengths:

- S1: The paper tackles a real and specific benchmark maintenance problem in multimodal evaluation. The motivation is well framed that static multimodal benchmarks become stale, contaminated, and costly to update. The paper treats benchmark evolution as a controlled data construction problem rather than merely generating more questions.

- S2: The executable verification idea is technically meaningful. Instead of using a VLM to directly re-judge the same image-question pair, the framework requires the generator to output a solution plan and verifies correctness through tool execution, with the controller operating only over textual intermediate outputs

- S3: The paper includes cross-version correlation and contamination analysis, which supports the benchmark construction claim.

### Weaknesses:

- W1: The feedback controller judges whether an image-text pair matches the task purpose and visual patterns, producing a binary decision, confidence score, and diagnostic message. But the paper does not specify the prompt, thresholding rule, batch frequency, stopping criterion, acceptance rate, or how the diagnostic signal actually changes future retrieval behavior. Equation (3) defines outputs, but the mechanism linking feedback to acquisition is described only qualitatively. This makes it hard to assess whether the controller is a minor filter or a major driver of the final dataset.

- W2: The paper claims that visual patterns and feedback preserve distributional alignment, but the main evidence is FID on CLIP embeddings plus PCA-KL asymmetry. These are coarse feature-space statistics and do not directly show preservation of task semantics or answer-type distribution. The interpretation of positive versus negative $\Delta KL$ in Table 4 is somewhat heuristic. The paper would be more convincing if it showed per-task distribution preservation for answer formats, OCR density, object counts, relation types, or other benchmark-native properties derived from the task hierarchy.

- W3: The paper adapts PaCoST by comparing confidence on original versus paraphrased questions using a single model, Qwen2.5-VL-7B-Instruct. This measures text-side paraphrase sensitivity more than image-side contamination. Since the benchmark is multimodal, leakage can occur through the image itself, metadata, or web content. The reported effect size difference between MMBench and MMBench-Live is directionally encouraging, but the absolute effect sizes are small, and the analysis does not disentangle whether improvements come from newer images, different text style, or reduced benchmark recognizability.

- W4: For the appendix examples, several items appear easy or rely on generic world knowledge rather than demanding benchmark-specific multimodal reasoning. For instance, some “identity” or “social relation” questions infer occupation or relationship from stereotypical visual cues, while some “future prediction” items hinge on likely next action rather than a grounded, uniquely verifiable outcome. This weakens the claim that executable verification fully controls semantic quality, because the verification may confirm internally consistent reasoning without guaranteeing that the task is discriminative or free from social prior shortcuts. The examples on pages 20, 22, 26, and 30 particularly illustrate this concern.

- W5: The paper reports a 96.06% average correctness rate by sampling 10% of images per task and repeating the process three times. However, it is unclear whether correctness includes only answer correctness or also task validity, ambiguity, and faithfulness of the solution plan. Given that the benchmark is built around executable reasoning, auditing only the final QA correctness may miss cases where the plan is brittle, overfitted to metadata, or semantically misaligned with the intended capability.

---

> ### Author Rebuttal · Authors · 2026-03-31
>
> We thank the reviewer for the careful reading and constructive feedback. Below, we respond to each point in turn and clarify the corresponding aspects of our method and analysis.
>
> 1. Feedback Controller Design and Influence on Data Acquisition (to w1)
>
> Thank you for this important comment. In our implementation, the feedback controller is a lightweight online correction module rather than the main driver of dataset construction. It takes the task description and task-related visual patterns as input, and outputs a binary decision with 1–3 concise, visually grounded reasons. After each retrieval round, we evaluate the top 10 returned images: if more than 70% are accepted, we keep the current query; otherwise, we regenerate the query using the rejection reasons. This process runs for at most three iterations. Once the acceptance condition is met or the maximum number of iterations is reached, the controller stops updating retrieval and only serves as a filter for subsequent candidates.
>
> 2. More Statistics on Distribution Alignment (to w2)
>
> We appreciate this suggestion. To complement these coarse statistics, we note two additional controls in MMBench-Live: The sample proportion across tasks is explicitly balanced for fairness and comparability. And since MMBench consists entirely of multiple-choice questions, MMBench-Live also uses only multiple-choice QA pairs with the same number of answer options, so the answer format is controlled by design.
> We further analyze one representative subtask, spatial relationship, where task-relevant labels can be reliably recovered by LLM from QA pairs. We compare the label inventory and frequency distribution. MMBench contains 18 label types, with top-5 labels north, east, west, northwest, south; MMBench-Live contains 36 label types and has top-5 labels north, northeast, east, south, west. The strong overlap in high-frequency labels suggests that the updated benchmark preserves the core semantic structure of this subtask. We attribute this to task-related visual patterns and our distribution-consistent update strategy, which help the agent capture recurrent structures such as the prevalence of map-based instances. We will include these analyses in the revision.
>
> 3. Clarification on Contamination Analysis (to w3)
>
> We appreciate this point. Our PaCoST-based analysis is intended as a contamination-related proxy rather than a complete diagnosis of multimodal leakage. Because PaCoST compares confidence on original versus paraphrased questions, it is more sensitive to text-side memorization and does not explicitly disentangle image-level leakage, metadata overlap, or web-content exposure. Our claim is therefore relative rather than absolute: under the same protocol, MMBench-Live shows a weaker contamination-related signal than MMBench, reflected by the smaller mean confidence gap and effect size. This trend is also consistent with our design, which uses newly collected real-world images, temporal isolation, and MMBench-style QA generation. We will clarify in the revision that this result should be interpreted as supportive evidence, while a more controlled disentanglement of image freshness, text style, and recognizability is left for future work.
>
> 4. On Executable Verification and Its Scope (to w4)
>
> We appreciate this important observation. Executable verification is intended to improve reliability and reduce hallucination, rather than to guarantee that every example is maximally discriminative or fully free from social-prior shortcuts. Even for tasks such as identity or social relations, the answer still needs to be grounded in observable evidence and checked through the executable process, instead of being accepted from unconstrained model judgment. In this sense, executable verification is effective for most tasks in ensuring visual grounding and answer correctness, while not by itself guaranteeing full resistance to shortcut solutions. In future work, we will introduce more comprehensive filtering and verification mechanisms.
>
> 5. On Correctness Evaluation (to w5)
>
> We appreciate this comment. The reported 96.06% correctness rate evaluates only the correctness of the final QA pairs, rather than task validity, ambiguity, or solution-plan faithfulness. This is consistent with the role of executable verification in our framework, which is to improve instance reliability and reduce hallucination by ensuring that the final answer is supported by an executable checking process, rather than imposing a complex reasoning chain on every QA pair. This is also consistent with the scope of MMBench, where many items focus on core visual understanding and only a subset require more explicit reasoning. We will clarify this scope in the revision and consider more fine-grained manual inspection of plan brittleness, metadata overfitting, and semantic misalignment in future work.

---

> > ### Author Rebuttal · Reviewer_a7FG · 2026-04-04
> >
> > Thank authors for the responses. I would appreciate the claimed updates to the manuscript.

---

### Decision · Program_Chairs · 2026-04-30

**Decision:**

Accept (regular)

**Comment:**

This paper introduces MMBench-Live, a multi-agent-driven dynamic multimodal benchmark that supports continuous updates without human intervention. It is maintained through an end-to-end automated pipeline that integrates structured benchmark specification, real-time data acquisition, and verifiable question–answer generation, enabling scalable, live, and low-cost benchmark updates.

This paper received two Weak Accept ratings (Reviewer a7FG, Reviewer YaYB) and one Weak Reject rating (Reviewer D7x3). After the rebuttal, Reviewer D7x3 posted the remaining concerns, and the authors provided further clarification. Although not all concerns were fully resolved, the work does not have any crucial weakness. Therefore, the final rating is Weak Accept.